# Systematic identification of intron retention associated variants from massive publicly available transcriptome sequencing data

Yuichi Shiraishi [1,4] ✉, Ai Okada[1,4], Kenichi Chiba[1], Asuka Kawachi[2], Ikuko Omori[2], Raúl Nicolás Mateos[1], Naoko Iida[1], Hirofumi Yamauchi[2], Kenjiro Kosaki[3] & Akihide Yoshimi [2]

Many disease-associated genomic variants disrupt gene function through abnormal splicing. With the advancement of genomic medicine, identifying disease-associated splicing associated variants has become more important than ever. Most bioinformatics approaches to detect splicing associated variants require both genome and transcriptomic data. However, there are not many datasets where both of them are available. In this study, we develop a methodology to detect genomic variants that cause splicing changes (more specifically, intron retention), using transcriptome sequencing data alone. After evaluating its sensitivity and precision, we apply it to 230,988 transcriptome sequencing data from the publicly available repository and identified 27,049 intron retention associated variants (IRAVs). In addition, by exploring positional relationships with variants registered in existing disease databases, we extract 3,000 putative disease-associated IRAVs, which range from cancer drivers to variants linked with autosomal recessive disorders. The in-silico screening framework demonstrates the possibility of near-automatically acquiring medical knowledge, making the most of massively accumulated publicly available sequencing data. Collections of IRAVs identified in this study are available through IRAVDB (https://iravdb.io/).

During the continued innovation in sequencing technology, the effectiveness of large-scale genome analysis has been thoroughly validated and widely recognized. Nowadays, national-scale genome projects have been moving forward worldwide, and genome analysis is expected to revolutionize the medical system. It is more important than ever to identify disease-associated variants from a vast list of mutations obtained by sequencing patients. However, there still remain many challenges for interpreting the effect of each genomic variant, especially those that contribute to disease by a different mechanism than amino-acid substitutions.

One important class of pathogenic variants is those causing abnormal splicing changes, typically by damaging existing splicing motifs or creating novel splicing motifs. They may comprise 15–60% of human disease variants[1,2]. There have been many attempts to catalog these splicing associated variants. One approach is to resort to machine learning-based methods[3]. However, such approaches are still in their infancy in terms of precision and recall. In addition, predicting the consequence of splicing changes, which is often vital for evaluating the pathogenicity of variants[4], is even more challenging. Another approach is to perform genome and transcriptome analysis and

[1]Division of Genome Analysis Platform Development, National Cancer Center Research Institute, Tokyo, Japan. [2]Cancer RNA Research Unit, National Cancer Center Research Institute, Tokyo, Japan. [3]Center for Medical Genetics, Keio University School of Medicine, Tokyo, Japan. [4]These authors contributed equally: Yuichi Shiraishi, Ai Okada. ✉e-mail: yuishira@ncc.go.jp

identify pairs of genome variants and the corresponding splicing changes through some statistical methods[5–8]. However, this approach needs a dataset where both genome and transcriptome data are provided, which is not very common.

Here, to make the most of massive collections of transcriptome data available in public sequence repositories[9,10], we provide a approach, IRAVNet (https://github.com/friend1ws/iravnet), which can identify genomic variants that cause specific types of splicing aberrations, intron retention, by using transcriptome sequencing data alone. Another advantage of IRAVNet is that, unlike previous methods that assess the association between genomic variant status and the amount of splicing changes, this pipeline can be run on single transcriptome sequencing data without gathering or specifying a set of data. This makes a systematic screening of public repositories much easier. We confirmed that intron retention associated variants (IRAVs) can be detected with high accuracy as well as a certain degree of sensitivity based on transcriptome sequencing data. Using this method, we performed a comprehensive screening of intron retention variants mainly using massive public transcriptome sequencing data registered in Sequence Read Archive (SRA). Also, we have prepared a web-based portal site, IRAVDB (https://iravdb.io/), where users can view various information about the IRAVs of interest, such as positional relationships with registered pathogenic variants and a list of the SRA samples with those variants.

## Results

### Method overview

When a variant is a direct cause of intron retention by disrupting existing splicing donor/acceptor motifs, we can observe mismatched bases in most of the retained short-reads at the exon–intron boundary (Fig. 1a and Supplementary Fig. 1). We exploited this phenomenon to develop an algorithm that can precisely identify IRAVs. Briefly, IRAV-Net first lists up putative variants supported by three or more short reads around the exon–intron boundaries (three exonic and six intronic bases for splice donor sites, and six intronic and one exonic base for splice acceptor sites). Next, IRAVNet only keeps the candidates whose variants are specifically supported by the intron-retention

supporting reads and not by normally spliced reads (Supplementary Fig. 2). Finally, IRAVNet removes the potential artifacts such as those presumably produced by alignment errors around exon-intron boundaries. We also removed common variants (those whose allele frequencies are greater than 0.01 by gnomAD database[11]) to focus on variants having a significant effect on disease while keeping the false positive rate low. See the Method section for a more detailed description.

### Application on TCGA dataset

To test the effectiveness of this approach, we conducted a preliminary analysis using 11,312 transcriptome sequencing data from The Cancer Genome Atlas (TCGA). In total, 2693 IRAVs were identified after merging the variants with the same position and substitution found in multiple samples (Supplementary Data 1). To investigate whether the predicted IRAVs identified from transcriptome by IRAVNet are truly genomic variants or not, we developed a framework for assessing the genomic mutation status of IRAVs using paired exome sequencing data. We classified the IRAVs into somatic, germline, somatic or germline, ambiguous, or false positive by exploring the number of supporting reads and sequencing depths at the positions of IRAVs for the 2967 corresponding pairs of tumor and matched control exome sequencing data (Fig. 2a). This revealed that the ratio of false positives in terms of genomic mutation status was estimated to be as low as 0.96% (Fig. 2b). Next, we evaluated the amount of intron retention for the samples with IRAVs compared to others. We confirmed that, in most cases, significantly higher ratios of intron retention were observed specifically in the samples having IRAVs (Fig. 2c). To evaluate the sensitivity of the proposed approach, we performed a comparison with our previous approach, SAVNet[5,12] which collects somatic splicing associated variants making use of paired genome and transcriptome data. IRAVNet, which just utilizes transcriptome data, detected about 43.1% (1032/2393) of intron retention causing somatic variants identified by SAVNet. In addition, out of the 1291 variants classified as "somatic" by the above procedure (Fig. 2a), IRAVNet identified 331 "new" variants, in the sense that they were not identified by SAVNet or identified as associated with other types of splicing changes than

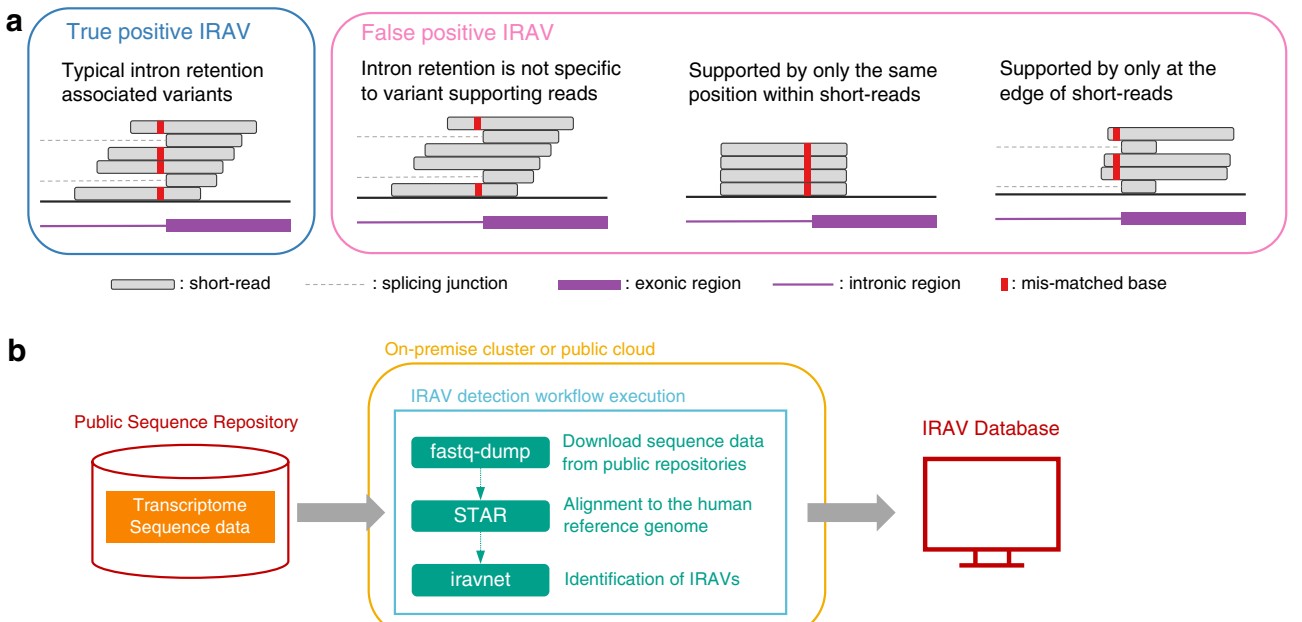

**Fig. 1 | Schematics of identification intron retention associated variants.**
**a** Examples of transcriptome sequencing alignment around variant causing intron retention, as well as common patterns of false positives. **b** Overview of the proposed framework for detecting intron retention associated variants from raw

sequencing data registered in Sequencing Read Archive. Downloaded sequence data is processed in on-premise or public cloud computing environment and identified IRAVs are transferred to the IRAV database and provided to the community.

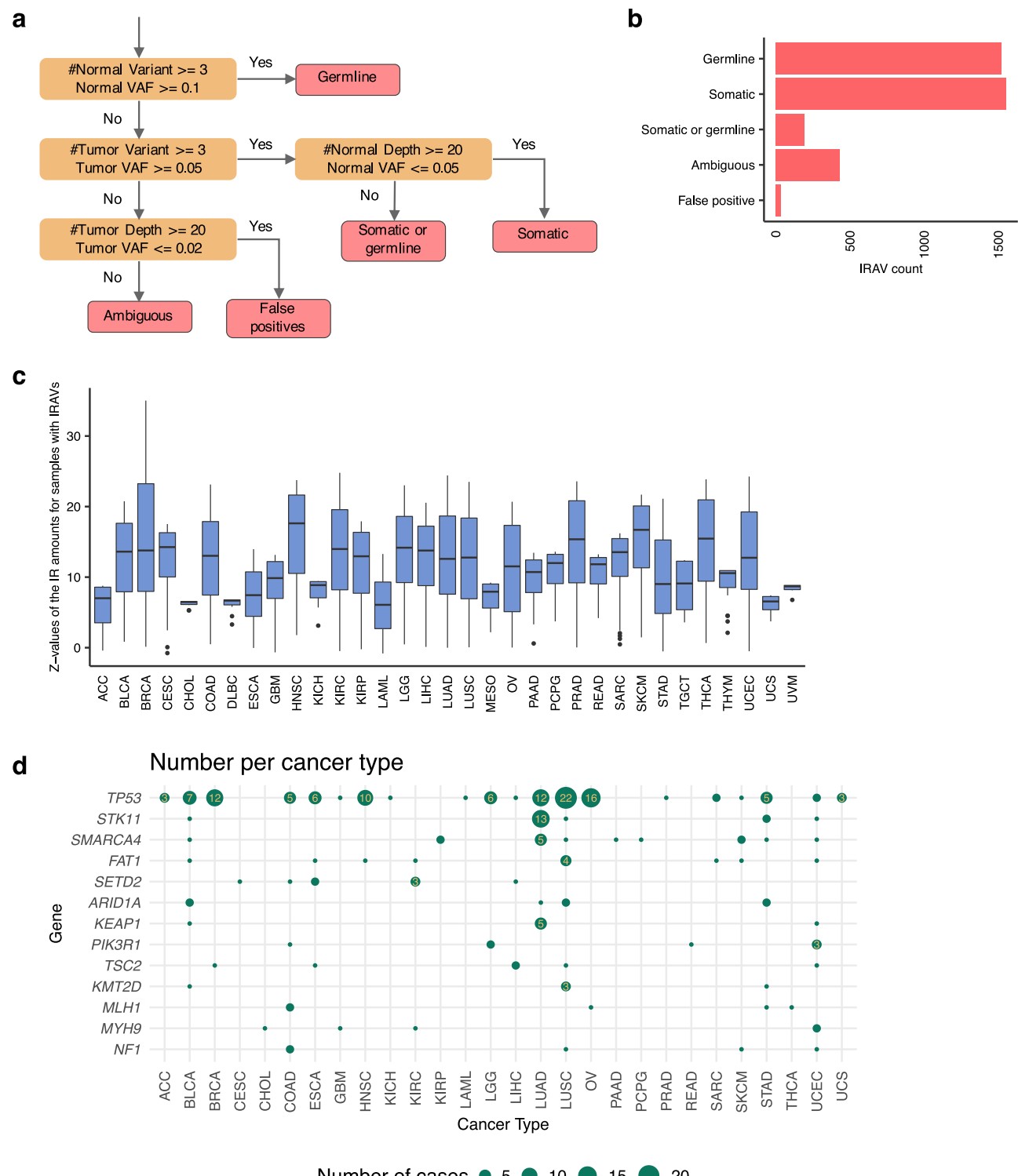

**Fig. 2 | Assessment of the IRAVNet approach using TCGA transcriptome and exome sequencing data. a** The flowchart showing the classification of genome level mutation status for IRAVs. Here, according to sequencing depths, the variant counts (the numbers of intron retention reads), and variant allele frequencies (VAF, the ratios of intron retention reads to the total number of sequencing reads covering the corresponding exon-intron boundaries), the status are classified into "germline," "somatic," "somatic or germline," "ambiguous", and "false positive." **b** The number of IRAVs categorized by the inferred mutation status for the identified IRAVs determined by the above procedure. **c** The boxplot showing how the intron retention is specific to the samples with the IRAVs. The ends of the boxes indicate lower and upper quartiles; center line, median; whiskers, maximum and minimum values within 1.5 × IQR from the edges of the box, respectively. For each IRAV, the $Z$-value comparing the ratios of intron retention between samples with the IRAV and other samples in the same cancer type group is computed. We observed that most of the $Z$-values were above the reasonable threshold (>2), strongly suggesting that most IRAVs certainly generate intron retention. The information, such as the sample size for each box, is provided in the Source Data file. **d** Landscape of IRAVs in frequently altered cancer-related genes (total number ≥5) across cancer types. The point size indicates the number of affected samples. Genes are sorted by the total number of IRAVs in all cancer types. See also Supplementary Fig. 4.

intron retention (Supplementary Fig. 3). Therefore, IRAVNet achieves a certain level of sensitivity and a high rate of precision, even though it only uses transcriptome data. Furthermore, most of the IRAVs detected by this approach are thought to actually cause intron retention. IRAVNet was able to identify 321 distinct IRAVs that affect well-known cancer genes such as *TP53, STK11, SMARCA4, FAT1*, and *SETD2* (Fig. 2d, Supplementary Fig. 4, and Supplementary Data 2). In addition, strong enrichment of several genes on specific cancer types reflected well on the previous findings (e.g., high concentration of *STK11* and *KEAP1* in lung adenocarcinomas[13]). These results indicate that this approach can effectively catalog disease-associated variants.

### Application on GEUVADIS dataset

We further evaluated the IRAVNet approach using GEUVADIS RNA sequencing data[14]. Among 652 transcriptome sequencing data whose matched whole-genome sequencing data is available, we identified 68 distinct IRAVs (variants sharing the same position and substitution identified in multiple samples were counted as one). Most of the IRAVs (66/68) were also detected as genomic variants by standard variant detection pipeline on independent whole-genome sequencing data[15], confirming that these IRAVs are at least actual genomic-level mutations. To explore whether these IRAVs have certain effects on intron retention, we divided samples into IRAV positive and negative groups based on genotyping via whole-genome sequencing data and compared the amount of intron retention measured by several methods (IRFinder[16] and MAJIQ[17,18]) for each IRAV. When we measured the local intron retention ratio around the exon-intron boundary (IRFinder LocalIRRatio), the effect on intron retention was rather clear for most IRAVs, corroborating that these IRAVs were likely to induce intron retention ($Z$-value ≥2 and $p$-value ≤0.01 for 97.0% IRAVs, see Supplementary Figs. 5–8). When measuring via intron-wide sequencing depth (IRFiner IRRatio, MAJIQ), the effect becomes slightly weaker, possibly because the consequences of splicing causing variants and their appearance in transcriptome data are often complex. We also applied IRAVNet to downsampled transcriptome sequencing data to assess the robustness of IRAVNet approach, confirming a considerable amount of IRAVs could be reproduced (Supplementary Fig. 9a).

Next, we compared our approach with machine learning-based splicing effect prediction approaches (SpliceAI[3], MMSplice[19]). The variants identified by IRAV showed more substantial enrichment of intron retention events than those predicted by machine learning prediction methods (Supplementary Fig. 9b). We believe this is because IRAVNet approach directly observes the variants from transcriptome sequencing data and may have an advantage over purely predictive approaches based on the nucleotide sequences. In addition, splicing aberrations caused by genomic variants are heterogeneous (including exon skipping, alternative 5 and 3′ splice site), and current machine learning approaches have not been specifically trained to predict intron retention.

### A computational framework for applying Sequence Read Archive

Next, to obtain a more comprehensive list of IRAVs, we applied this approach to publicly available transcriptome sequencing data from SRA[20] (Fig. 1b). For this purpose, we developed a cloud-based platform utilizing Amazon Web Service (Supplementary Fig. 10) as well as on-premise computational clusters (Supplementary Fig. 11). For the cloud-based platform, we automated the entire process of analysis using serverless architecture; download the raw sequence data, alignment to the human reference genome, and perform IRAVNet to identify IRAVs. To make the analysis reproducible, we utilized a container orchestration framework. We made various efforts to keep the cloud usage cost down by choosing the region where the data is located (to reduce downloading time), selecting the optimal instance type for each

procedure, setting the right amount of block storage to be reserved, and usage of spot instances (Supplementary Fig. 12).

### Screening of intron retention associated variants using Sequencing Read Archive

We have analyzed 219,615 transcriptome sequencing data (counted based on run IDs) and integrated the result into that of TCGA data (Fig. 3a). In the SRA, there were many sequencing data with different run IDs that were actually derived from the same individuals as exemplified by frequently used cell lines with multiple experimental conditions. Therefore, to avoid double counting of IRAVs, we basically focused on "distinct IRAVs" in the following, in which those with the same genomic position and substitution patterns identified from multiple sequence data are counted as one.

We detected 27,049 distinct IRAVs in total. Of which, 25,849 were on coding genes defined by RefSeq, and 119, 24,001, and 1729 distinct IRAVs involved splices sites at 5′UTR, coding, and 3′UTR regions, respectively. Most of the IRAVs associated with coding regions were predicted to create premature termination codons (23,525, 98.0%), 23,332 (99.2%) of which were inferred to be nonsense-mediated decay (NMD) sensitive by the 50nt rules[21] (premature termination codons are located before the 50bp upstream of the last exon-exon junction, see Method section for details). In the following, we focused on these NMD-sensitive IRAVs (21,584 SNVs and 1748 indels in 7813 RefSeq coding genes) because they are plausible to be associated with loss-of-function[11]. In all, 16,976 IRAVs were located at splice donor sites, whereas 6356 IRAVs disrupted splice acceptor sites. 10,414 (44.6%) IRAVs were those that did not involve GT-AG essential splice sites (Fig. 3b). The distribution of substitution patterns of identified IRAVs at the splice donor and acceptor sites show similar patterns for splicing associated variants with the previous study[5], where many were found at GT-AG essential splices sites followed by the last exonic and the 5th intronic bases of splice donor sites (Fig. 3c). Furthermore, splicing donor motifs grouped by the relative positions of IRAVs were separated into left-handed and right-handed motifs, where higher entropy masses were placed in the exon and in the intron, respectively[22] (Fig. 3d).

### Pathogenicity of intron retention associated variants obtained from Sequence Read Archive

Next, to search for IRAVs that are considered to be linked to disease, we took an approach of extending existing knowledge related to disease and we divided variants into 5 tiers based on the positional relationships with the known pathogenic variants registered in ClinVar[23] (Fig. 4a and Supplementary Data 3). 354 IRAVs sharing the same positions and substitution patterns with already registered pathogenic/likely pathogenic (P/LP) splicing variants were defined as Tier 1. In total, 694 IRAVs sharing the same splice sites with registered P/LP splice-site variants were classified as Tier 2. 866 IRAVs for which P/LP truncating variants are seen within 30 bp from the exon-intron boundary of the IRAVs were categorized as Tier 3. In all, 1086 NMD-sensitive IRAVs that affect genes with at least one registered P/LP truncating variant were designated as Tier4, and other variants were set to Tier 5. Here, Tier 1 to Tier 4 IRAVs were considered to be putative pathogenic IRAVs (ppIRAVs) and will be discussed in the following. The number of ppIRAVs identified showed an almost linear increase with the number of transcriptome sequence data analyzed and has not yet reached saturation. As the data accumulate at an accelerated rate, much more ppIRAVs are expected to be found via further analysis (Fig. 4b).

### Putative pathogenic intron retention associated variants affecting disease-related genes

The transcriptome sequencing data from Sequencing Read Archive includes a number of cancer cells. In total with TCGA RNA sequencing data, 636 ppIRAVs (Tier1: 123, Tier2: 207, Tier3: 145, Tier4: 161) were

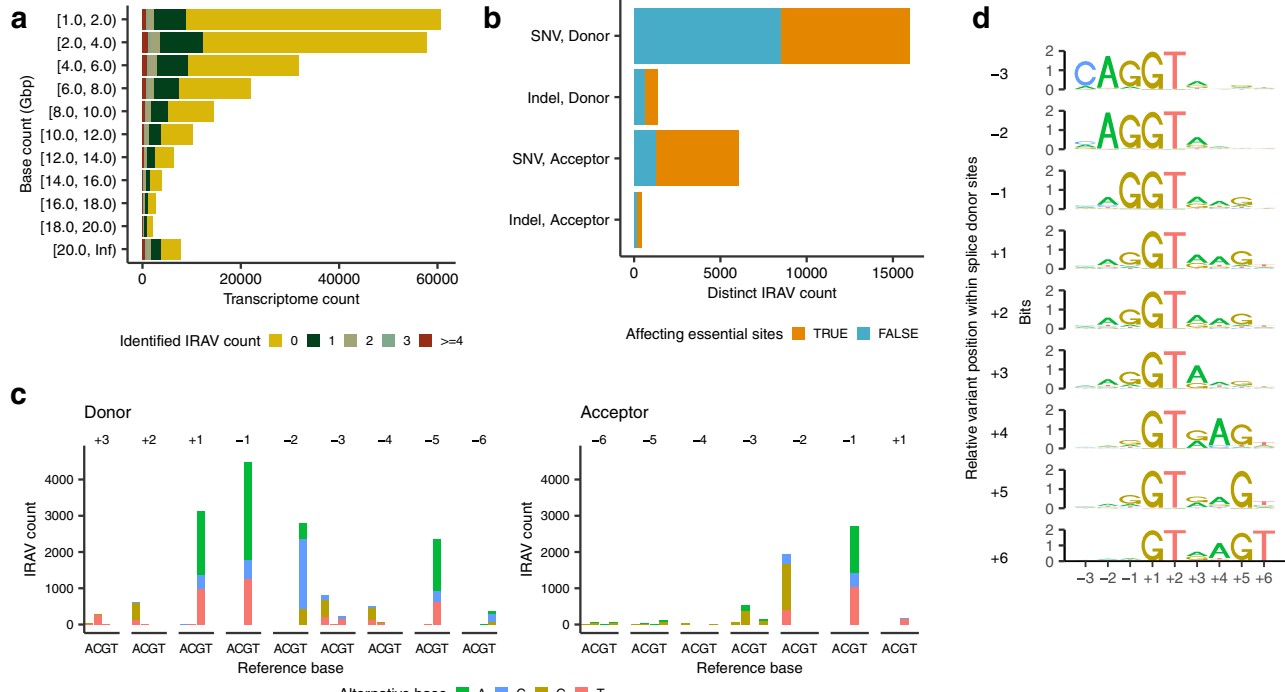

**Fig. 3 | Overviews of IRAVs identified from Sequencing Read Archive.**
**a** Frequencies of transcriptome sequencing data analyzed binned by the amount of base counts. Transcriptome data were also grouped by the number of detected IRAVs. For example, in the rows [1.0, 2.0), zero, one, two, three, and four or more IRAVs were identified in 51,793, 6,600, 1,187, 419, and 659 transcriptome sequence data, respectively, whose base count are ≥1.0 Gbp and <2.0 Gbp. **b** The number of distinct IRAVs affecting or not affecting essential splice-site (GT-AG), stratified by

(1) donor or acceptor and (2) SNVs or indels. **c** Base substitution patterns of IRAVs at each exonic and intronic position of splice donor and acceptor sites. Different colors are used to display different types of alternative bases. The x-axes represent different reference bases, and the y-axes represent the numbers of variants. **d** Sequence motifs of splicing donor sites for IRAVs classified by relative variant position to exon-intron boundaries.

---

those affecting cancer-related genes[24] (Fig. 4c). The fact that most of them (490, 77.0%) were only identified from SRA (Fig. 4d) indicates the usefulness of SRA, which is overwhelmingly superior in number, compared to TCGA data. Most of the cancer-related IRAVs identified from SRA were considered to come from cancer transcriptome data and to be somatic variants from the fact that their population allele frequencies (via gnomAD) were generally very low. Genes with ≥ 15 ppIRAVs were *TP53* (53 ppIRAVs), *MYH9* (26), *NF1* (24), *SMARCA4* (23), *STK11* (20), *RB1* (19), *FLNA* (17), *TSC2* (17), *BCOR* (15) (Fig. 5a, c, d and Supplementary Fig. 14). Generally, cancer-related ppIRAVs tended to concentrate on specific splice-sites, and 36 splice-sites had three or more ppIRAVs. In extreme cases, 19 ppIRAVs were concentrated at the 4th exon splice donor site of *TP53*, of which nine ppIRAVs have not been reported in ClinVar (Fig. 5b). Intron retention caused by the IRAVs at this site was shown to be associated with overexpression of Δ133p53 transcript[25], which may promote cancer cell invasion[26]. Six ppIRAVs were identified in the splice donor site of the 6th exon in *YY1AP1* gene, whose relationships with Grange syndrome[27] and hepatocellular carcinoma[28] have been reported. Other hotspot splice-sites included the splice donor site of *TP53* exon 9 (6 ppIRAVs), the splice donor site of *SMARCA4* exon 34 (5), and the splice acceptor site of *STK11* exon 7 (5).

Other recurrent ppIRAVs were detected in a number of genes known to be associated with disease such as *MYH9* (26 ppIRAVs), *PHKB* (16), *COL7A1* (14), and so on (Fig. 4c and Supplementary Fig. 15). *MYH9* has been known to cause an autosomal-dominant disease called MYH9-related disorder (MYH9-RD), characterized by large platelets and thrombocytopenia as well as increased risk of progressive nephropathy, sensorineural deafness, pre-senile cataract, and aberration of liver enzymes[29,30]. Moreover, several studies using animal models suggested that *MYH9* may act as a tumor suppressor and

inactivation of *MYH9* was related to the development of squamous cell carcinomas[31] and invasive lobular breast carcinomas[32]. *PHKB* is associated with the autosomal recessive type of glycogen storage disease type IX. Many truncating and splicing variants have been reported in ClinVar and we have identified IRAVs near them as well as other locations. Mutations in the *COL7A1* gene cause dystrophic epidermolysis bullosa, leading to subepidermal blistering and mucocutaneous fragility[33]. The severity of the disease varies depending on the type of mutation, and the degree to which the ppIRAVs cause intron retention may have a different impact on this disorder. In addition, 678 ppIRAVs (Tier1: 108, Tier2: 243, Tier3: 178, Tier4: 149) were those affecting haploinsufficient genes[34], and 250 ppIRAVs (Tier1: 79, Tier2: 108, Tier3: 34, Tier4: 29) were identified in ACMG genes list version 3.0[35] (Supplementary Fig. 13).

Collectively, our approach of screening ppIRAVs with massive transcriptome data can still greatly improve the current knowledge of variants causing splicing changes even for well-known disease-related genes.

**Putative intron retention associated variants related to drug response**

Loss of function mutations exemplified by variants causing splicing changes occasionally provides important information on the potential safety and effect of drugs[36]. We also identified IRAVs related to drug response in the same manner as identifying ppIRAVs: we compared positional relationships between variants reported to affect a drug response registered in ClinVar. We identified 13 IRAVs (Tier1: 2, Tier2: 2, Tier4: 9) predicted to be related to drug response (Supplementary Data 4). Eleven distinct IRAVs were identified in *DPYD*, mutations of which are implicated to increased risk of toxicity in cancer patients receiving 5-fluorouracil chemotherapy[37]. One variant, c.1905+1G>A, is

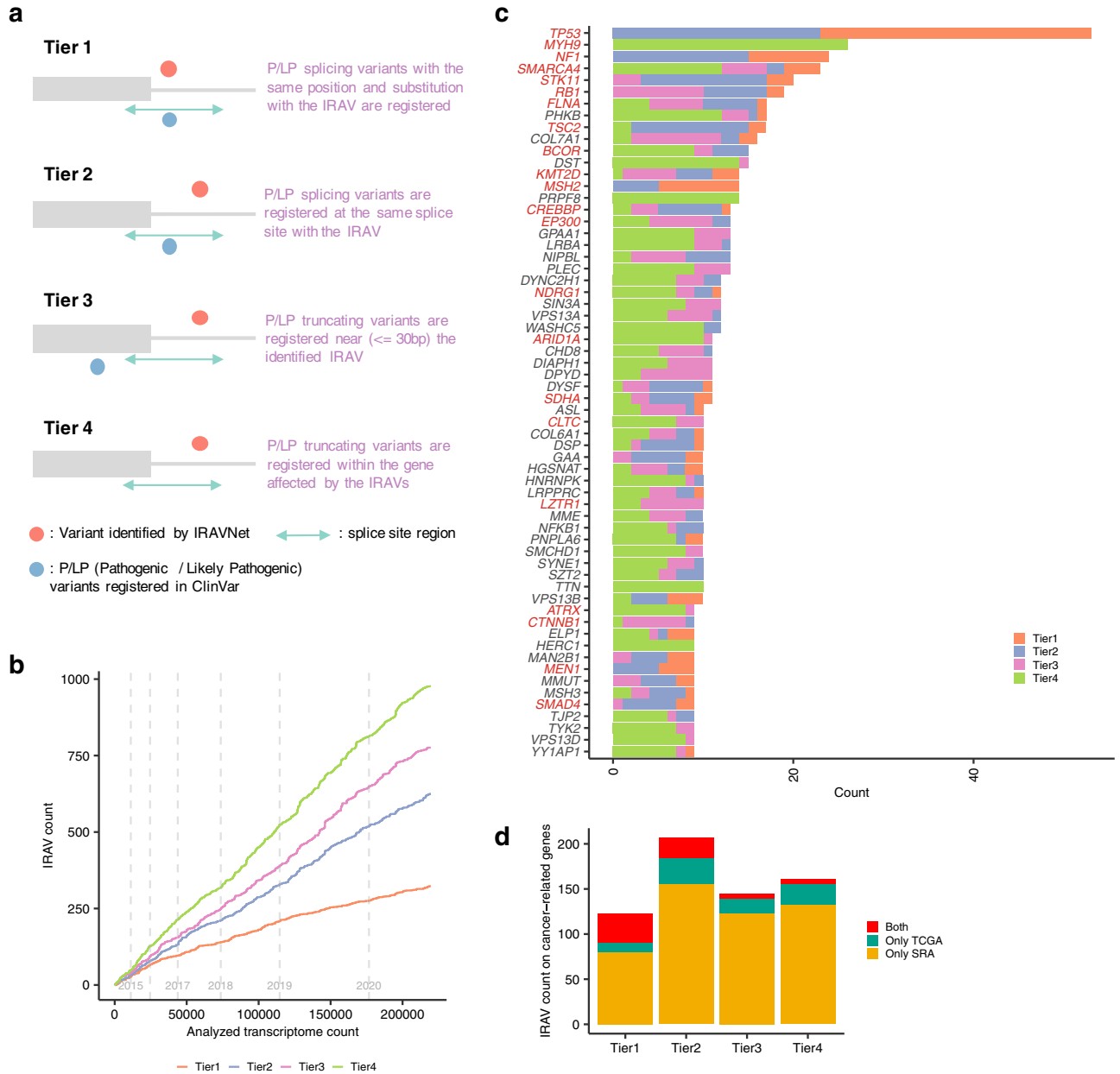

**Fig. 4 | The landscape of putative pathogenic IRAVs. a** A conceptual diagram for the classification of disease-related IRAVs in four tiers. The ranks of putative pathogenic IRAVs are determined by comparison with the positional relationships with the registered pathogenic/likely pathogenic (P/LP) variants in ClinVar. **b** The saturation analysis of putative pathogenic IRAVs via using Sequencing Read Archive. Transcriptome data were sorted by registration date, and the cumulative number of putative pathogenic IRAVs against the number of the analyzed transcriptome is shown. Gray vertical lines indicate the registration year changes. **c** The landscape of putative pathogenic IRAVs for frequently altered genes (total number ≥10). Genes were sorted by the total number of distinct variants, and known cancer-related genes are shown in red. **d** The number of distinct putative pathogenic IRAVs affecting cancer-related genes for each rank identified from Sequencing Read Archive, TCGA, and both.

relatively frequent (allele frequency: 0.0047), well characterized[38], and "reviewed by expert panel" status in ClinVar. Although the other nine IRAVs were very rare (≤0.0001 allele frequency), they are, in aggregate, potentially very important when administering medication. Other genes included *CYP2C19*, which is known to influence the enzyme activity in the metabolic pathway of drugs[39].

### Relatively common pathogenic intron retention associated variants implicated in genetic disorders

Even though we focused on rare variants (≤0.01 allele frequency), 178 ppIRAVs (Tier1: 52, Tier2: 35, Tier3: 35, Tier4: 55) were relatively common (≥0.0001 allele frequency). Among them, 18 ppIRAVs

(Tier1: 8, Tier2: 5, Tier3: 5, Tier4: 0) affected cancer predisposition genes[40] and 12 ppIRAVs (Tier1: 2, Tier2: 8, Tier3: 0, Tier4: 2) were located at ACMG genes list version 3.0[35]. They are considered to be germline variants that work as factors for genetic diseases and might be a good candidate for a drug target. Many of them have been registered in ClinVar as "Benign," "Likely benign," "Conflicting interpretations of pathogenicity," or "Uncertain significance" partly because the effects of splicing are unclear. Using transcriptome sequence data collected from multiple tissues (Genotype-Tissue Expression project[41]), we have confirmed that individuals with ppIRAVs tend to show specifically high intron retention ratios at corresponding exon-intron boundaries in multiple tissues (Fig. 6a

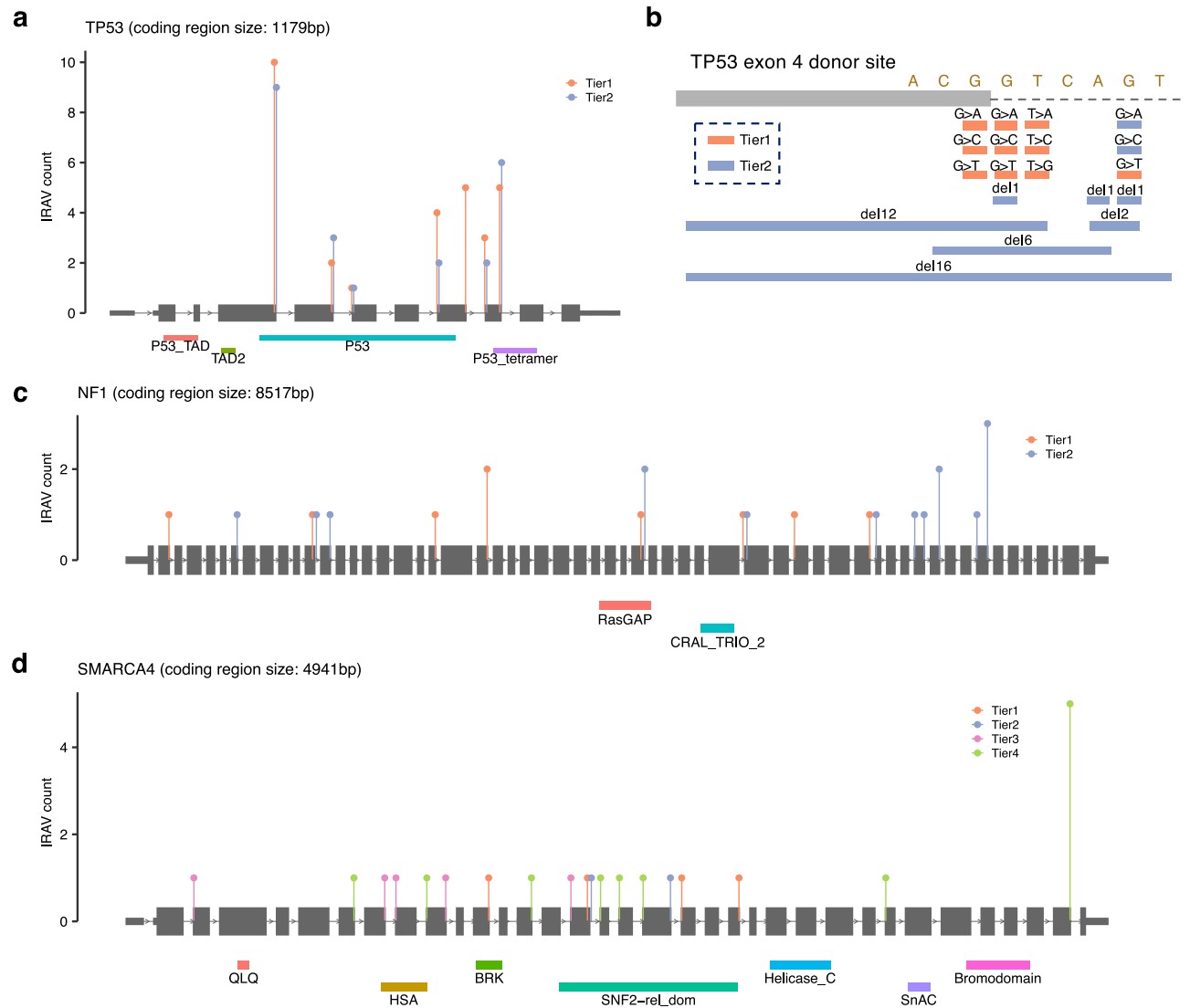

**Fig. 5 | Distribution of IRAVs in genes with frequent IRAVs. a, c, d** Frequencies of putative pathogenic IRAVs for each pathogenic tier at each splice-site are shown in **a** *TP53*, **c** *NF1*, and **d** *SMARCA4*. **b** The catalog of IRAVs at the *TP53* exon 4 donor site identified in this study. All substitution patterns at the essential splice site, the last exonic base, and the 5th intronic base were covered, as well as several deletions.

and Supplementary Figs. 16 and 17). Examples of relatively frequent ppIRAVs included the c.1473+5G>T variant at the 9th exon splice donor site of *P3H1*, linked with osteogenesis imperfecta. This variant was observed in 405/152,156 (0.2662%) alleles (392/41,440 (0.9459%) for the African ancestry) in gnomAD database v3.1.1 and has not been reported as pathogenic. We confirmed by mini-gene assays that this variant induces strong intron 9 retention (Fig. 6b). Given that this variant may cause intron 9 retention in multiple tissues and that a pathogenic variant was reported at the same splice site (c.1473+1G>T)[42], this variant may have some pathogenicity properties. The next example is the c.424+5G>A variant at the 8th exon splice donor site of *SMAD4*. Although this variant has an even smaller allele frequency (19/151,834, (0.0125%)), it has been reported in several cancer-cohort studies[43–45]. Previous studies have described this variant as a variant of uncertainty significance even though they pointed out the possibility that it causes some kind of abnormal splicing. However, intron retention verified by our mini-gene assay implied some kind of pathogenicity of this variant (Fig. 6b). Overall, our approach provides an effective screening method for cataloging major genomic variants that are responsible for genetic disease through abnormal splicing.

## Discussion

We have proposed a framework for screening pathogenic variants through abnormal splicing, making the most of the vast amount of transcriptome data available in public repositories. Our methodology identifies not only many previously detected pathogenic variants but also a vast amount of not described ones. In addition, our saturation analysis demonstrated that more and more mutations would be identifiable by keeping applying this methodology as the sequence data accumulate in the repository. Furthermore, the correspondence table between the variants and sample IDs will enable researchers to download the set of sequencing data with the variants of interest for further detailed analyses.

There are several caveats in the IRAVNet approach. We identified mismatch bases around the splice-site in the transcriptome sequence alignment and confirmed whether this is specific to intron-retention supporting reads. Therefore, this methodology detects events in which mutations cause intron retention in a splice site where intron retention does not commonly occur. However, it may not be sufficient to detect mutations that cause a quantitative change, such as a mutation that results in stronger intron retention at the innately intron retention-intolerant splice

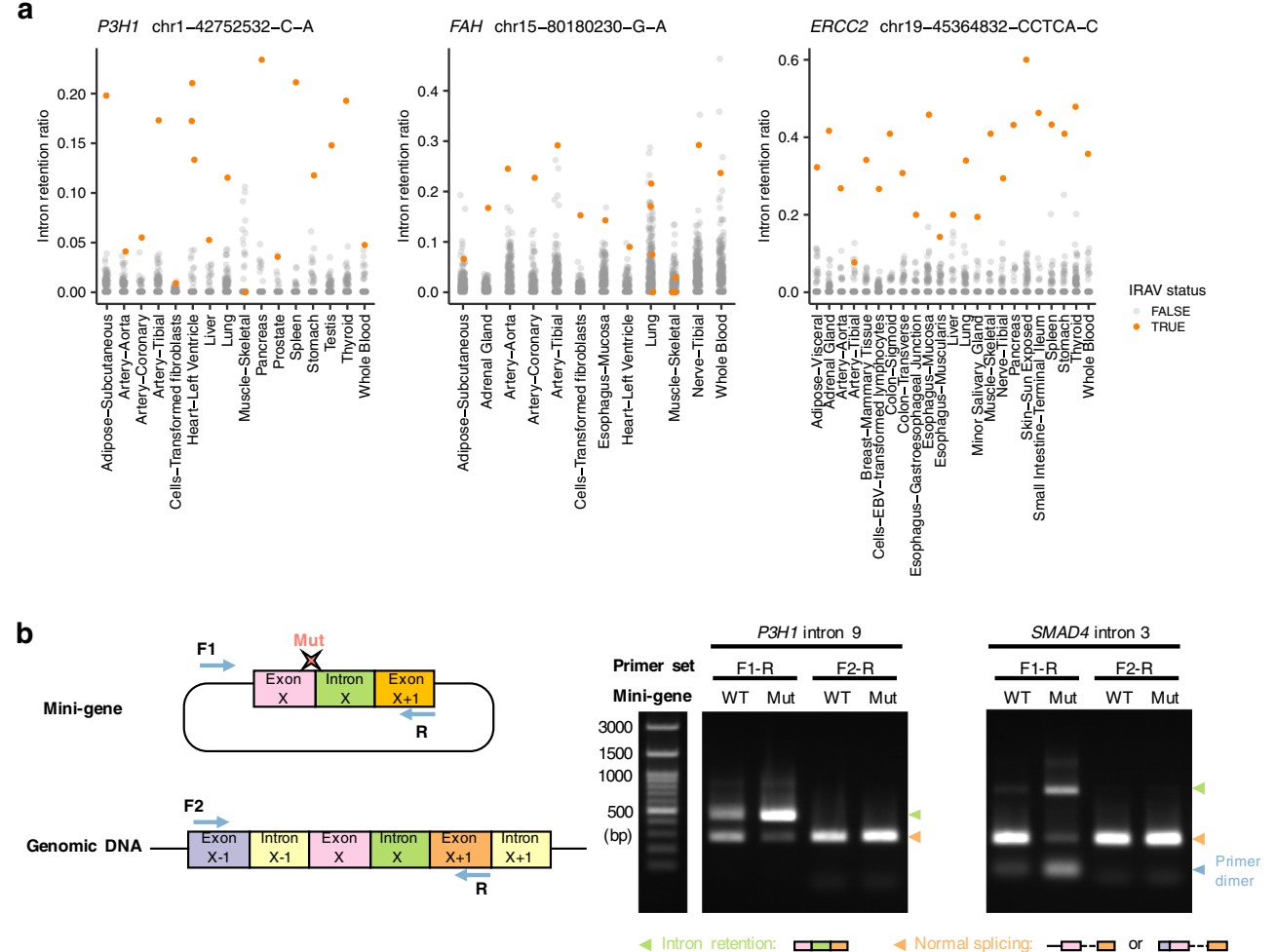

**Fig. 6 | Validation of relatively common IRAVs implicated in genetic disorders. a** Relative ratios of corresponding intron retention for samples with (orange) and without (gray) the IRAVs across tissues measured using GTEx transcriptome data. The p-values measuring the differences of the intron retention ratios between samples with and without IRAVs via the one-sided Wilcoxon rank-sum test for each tissue and integration by Fisher's method were $1.82 \times 10^{-42}$, $2.91 \times 10^{-7}$, and $1.22 \times 10^{-69}$, respectively from the left panel to the right. **b** In vitro splicing analyses using wildtype (WT) or mutant (Mut) minigene constructs (left) showing intron retention introduced by IRAVs at *P3H1* intron 9 and *SMAD4* intron 3. The experimental results were confirmed by three independent biological replicates, and representative results were shown.

site, where there are already some intron retentions. Other approaches relying on statistical associations would be necessary to identify these variants. Also, the prediction of NMD sensitivity could be improved. We utilized the 50nt rule for the prediction of NMD sensitivity as other genomic studies typically did. However, recent advances in this field have indicated the complex physiological regulation mechanism of NMD and the 50 nt rule may not be sufficient[21,46].

By utilizing the massive publicly available transcriptome sequencing data, attractive future works would be to collect other types of splicing associated variants than those causing intron retention by using transcriptome sequencing data alone (such as splice-site creating variants and mobile element insertions). However, several challenges remain to overcome, not only the detection of splicing associated variants itself. The Sequence Read Archive, which we mainly used in this paper, is not well organized in terms of metadata, making it difficult to perform association studies with some phenotypes. It may be helpful to devise and apply computational prediction methods for basic information such as race, tissue, and cancer status. Furthermore, the assessment of the pathogenicity of the splicing associated variants, while predicting the forms of consequent transcripts, will not be straightforward.

Previous approaches such as splicing QTL[14,47] evaluate the association between genomic variant status and the amount of splicing changes across samples. However, IRAVNet can be interpreted as evaluating the association differently: It examines the co-occurrence of mutations and splicing events (intron retention) across short reads. This can be achieved by drawing out the multi-layered information provided by the sequencing data (in this case, the presence or absence of both splicing changes and genetic mutations can be extracted). We believe that these ideas could be extended in another way. For example, the relationship between splicing and RNA modifications may be evaluated in a similar manner based on recent long-read sequencing technology.

With the accumulation of genome data via the implementation of genomic medicine, it will be increasingly important to acquire relevant knowledge for medical treatment and prevention. In principle, it is possible to build a system that automatically stores IRAVs by running IRAVNet in conjunction with data registration and upload. Still, there are many challenges such as reanalysis after updating the software, data harmonization, and storage issues. However, while providing an integrative solution to them, we believe that platforms that can efficiently and autonomously perform knowledge discovery will revolutionize the research and healthcare system.

## Methods

### A workflow for the discovery of intron retention associated variants

**Selection of Sequence Read Archive and TCGA samples.** Public Sequence Read Archive (SRA) samples were selected in a manner similar to the previous study[9]. We queried the SRA website (https://www.ncbi.nlm.nih.gov/sra) with the following search term: "platform illumina"[Properties] AND "strategy rna seq"[Properties] AND "human"[Organism] AND "cluster public"[Properties] AND "biomol rna"[Properties]. Then, we extracted samples whose base number is ≥1 billion bases, to secure sufficient sequence coverage for reliably detect mutations. There were a number of run data that could not be downloaded even after repeated trials probably due to some technical issue. Some of the downloaded sequence data had severe problems such as inconsistencies between two paired-end files, different lengths between sequence letters and base qualities, and so on). In addition, we removed run data whose number of IRAVs was extremely high due to potential DNA contamination and so on. For The Cancer Genome Atlas (TCGA) transcriptome data, we used all the available sequencing data at the Genomic Data Commons.

**Downloading Sequence Read Archive samples.** We used SRA Toolkit version 2.10.0. First, we performed prefetch command with the "----max-size 100000000" option to download SRA format file. Then, we executed fasterq-dump command with the option "-v --split-files."

**Alignment of RNA-seq data.** We used the GRCh38 based reference genome provided from Genomic Data Commons (https://gdc.cancer.gov/about-data/gdc-data-processing/gdc-reference-files). First, genome indexes were generated using STAR version 2.7.2b[48] with that reference genome and the release 31 GTF file (ftp://ftp.ebi.ac.uk/pub/databases/gencode/Gencode_human/release_31/gencode.v31.annotation.gtf.gz) and −sjdbOverhang 100 option. For each sample, alignment to the reference genomes was performed by the same version of STAR with the following options: --runThreadN 6 --outSAMtype BAM Unsorted --outSAMstrandField intronMotif --outSAMunmapped Within --outSJfilterCountUniqueMin 1 1 1 1 --outSJfilterCountTotalMin 1 1 1 1 --outSJfilterOverhangMin 12 12 12 12 --outSJfilterDistToOtherSJmin 0 0 0 0 --alignIntronMax 500000 --alignMatesGapMax 500000 --alignSJstitchMismatchNmax −1 −1 −1 −1 --chimSegmentMin 12 --chimJunctionOverhangMin 12. After the alignment, BAM files were sorted, and converted into CRAM format, and indexed using SAMtools version 1.9 (https://www.htslib.org/).

**Detection of intron retention associated variants from CRAM format files.** We focused on the known splice-site regions (from the 3rd exonic base to the 6th intronic base for splice donor sites, and from the 6th intronic base to the 1st exonic base) registered in RefSeq genes (https://hgdownload.cse.ucsc.edu/goldenPath/hg38/database/refGene.txt.gz) as previously described[5], and created the BED format file for the above splice-site regions. From these splice-site regions, we removed those completely included in the exonic regions of other genes or isoforms by RefSeq genes or GENCODE basic gene annotations. Also, using a panel of 742 control samples (collected from the TCGA transcriptome data), we filtered out splice-site regions where intron retention fraction is ≥ 0.05 in ≥ 8 control samples as previously described[5].

First, we piled up the CRAM file with the "samtools mpileup" command confining to the remaining 221,840 splice-site regions and identified putative variants with at least three variant supporting reads and ≥0.05 variant allele frequencies. Then, we checked the position of the variant in each supporting read and kept variants that are supported by at least three different positions (and at least one of which must be inside the 5 bases from the edges).

Next, for each variant remaining at this stage, we classified short reads around the variant by pair-wise alignment into:
- Splicing junction positive (negative): short reads that are normally spliced at the corresponding splice-site and do (not) support the target variant.
- Intron retention positive (negative): short reads that are not spliced and retained around the splice-site and do (not) support the target variant.

Let us denote the number of splicing junction positive, splicing junction negative, intron retention positive, and intron retention negative short reads as #SJ_Pos, #SJ_Neg, #IR_Pos, and #IR_Neg, respectively. We requested the following conditions:
- #IR_Pos ≥ 3.
- #IR_Pos / (#IR_Pos + #IR_Neg) ≥0.9 (to confirm that most intron retention reads include the target variant).
- #IR_Pos / (#SJ_Neg + #IR_Pos) ≥0.1 (to remove the variants with too low variant allele frequencies or too weak penetrance on the intron retention effect).

Also, at least one intron retention read needs to cover 25 intronic bases from the exon-intron boundary. Furthermore, to restrict reliable IRAVs, we requested that the MaxEntScore[49] be reduced by at least 2.5 due to the variants. Finally, using gnomAD version 3.0, we removed the variants with high allele frequencies (>0.01). The entire workflow is available at https://github.com/friend1ws/iravnet.

**Quantification of intron retention.** Intron retention was quantified by our in-house program (intron_retention_utils, https://github.com/friend1ws/intron_retention_utils) as in the previous study[5]. For each exon-intron boundary registered in RefSeq database, the number of presumed intron retention reads (those spanning ≥10 bp of both sides of the boundary), as well as that of normally spliced reads covering the last exonic base of the boundary, was counted.

### TCGA exome and transcriptome analysis

**Classification of intron retention associated variants using tumor and matched-control exome sequencing data.** For each transcriptome sequencing data having detected IRAVs, we first checked whether the corresponding tumor and matched control exome sequencing data is available in the Genomic Data Commons. Here, we focused on the sample type of "Primary Tumor," and "Primary Blood Derived Cancer - Peripheral Blood" (as well as "Metastatic" in cases when the cancer type is TCGA-SKCM) for tumor and "Solid Tissue Normal" and "Blood Derived Normal" for matched control. When both tumor and matched control were obtained, we downloaded part of the BAM file around the detected IRAVs using the BAM Slicing API. Then, we measured sequencing depths, the numbers of IRAV supporting reads, and the ratios of IRAV supporting reads for the IRAVs and classified them into "germline," "somatic," "somatic or germline," "ambiguous," and "false positive" according to the flowchart presented in Fig. 2a.

**Investigation of the specificity of the amount of intron retention.** The amount of intron retention was quantified by our in-house program (intron_retention_utils simple_count command) as described previously[5]. Briefly, for each exon−intron boundary, the number of intron retention reads (those covering ≥10 bp of both sides of the exon−intron boundary) and that of normally spliced reads covering the last exonic base of the exon−intron boundary were counted. For each detected IRAV, we measured the ratios of intron retention for the IRAV having sample and other samples with the same cancer type and quantified the Z-value comparing them.

**Comparison with SAVNet.** First, we confined the transcriptome data to those used in the previous study[5] for comparison. We collected the splicing associated variant identified as causing intron retention in that study and transformed the coordinates of these variants from those based on GRCh37 to GRCh38 using liftOver (https://genome-store.ucsc.edu/). Finally, we evaluated the overlap between the IRAVs identified in this study and them.

## GEUVADIS transcriptome analysis

**Processing Genotype data for GEUVADIS.** We downloaded processed VCF files[15] from http.www.s3://1000genomes/1000G_2504_high_coverage/working/20201028_3202_raw_GT_with_annot/. We extracted variants located within the splice-site regions specified by the BED files used in the IRAVNet approach and samples which have matched transcriptome using "bcftools view" and normalized these variants by "bcftools norm." As performed in IRAVNet workflow, we removed variants whose allele frequencies were more than 0.01 by gnomAD version 3.0 or that were not marked as PASS in the FILTER column, except for the variants identified by IRAVNet. Also, we filtered out those whose differential MaxEntScore by the variants are below 2.5 as performed in the IRAVNet procedure.

**Quantification of intron retention using IRFinder and MAJIQ and comparison of the amount of intron retention with and without IRAVs.** For all the genomic variants extracted in the previous subsection as well as IRAVs identified by IRAVNet, we quantified the amount of relevant intron retention using IRFinder version 1.3.1 and MAJIQ version 2.3. For using IRFinder, we first built a reference using "IRFinder BuildRef" command based on the Release 100 of Ensembl human genome GRCh38 gene annotations, including RNA.SpikeIn.ERCC.fasta.gz and Human_hg38_nonPolyA_ROI.bed files obtained from IRFinder repository (https://github.com/williamritchie/IRFinder). After downloading the transcriptome FASTQ files as described in the previous subsection, we quantified the intron retention via "IRFinder FastQ" command. We set the new indicator from the IRFinder result as LocalIRratioLeft = ExonToIntronReadsLeft/(SpliceLeft + ExonToIntronReadsLeft + 1), and LocalIRratioRight = ExonToIntronReadsRight/(SpliceRight + ExonToIntronReadsRight + 1). According to the positional relationship of the IRAV with the intron, one of the values of LocalIRratioLeft and LocalIRratioRight was set as the LocalIRratio. We also used the values of IRratio column as a measure for whole introns.

For using MAJIQ, we first downloaded a reference GFF3 file from https://ftp.ebi.ac.uk/pub/databases/gencode/Gencode_human/release_39/gencode.v39.annotation.gff3.gz. To quantify the amount of intron retention corresponding to the detected IRAVs, we inserted records corresponding to the new transcripts in intron retention assumed from the IRAVs were triggered. We performed "majiq build" command with "--irnbins 0.00001 --min-intronic-cov 0.00001 --annotated_ir_always" options, and quantified the relative abundance of splicing by "majiq psi" command with "--minreads 1 --minpos 1" options. We used the value of mean_psi_lsv_junction column as the quantification of intron retention.

For each IRAV, we divided the transcriptome samples into two groups according to the mutation status of IRAVs. We measured the difference between the two groups by Z-value via IRFinder (LocalIRratio and IRratio) and MAJIQ. Also, we performed a permutation test and obtained a p-value setting the statistics as the mean of the amount of intron retention for the positive groups. Furthermore, we calculated the difference in the median of each indicator of intron retention ratio between the two groups. We also performed MAJIQ HET to measure the difference between the two groups. For each IRAV, we performed the "majiq heterogen" command on the two groups based on the IRAV mutation status, and executed the "voila tsv" command with "--showall" option for each VOILA format file. Then, we calculated the difference between the values of columns whose suffixes are "true_median_psi" and "false_median_psi," and adopted the p-value from the "TNOM" column.

**Checking the reproducibility by sub-sampling.** To examine the reproducibility of IRAVNet approach, we subsample BAM files by "samtools view --subsample" command setting the subsample fraction to 0.1, 0.2, …, 0.9, and the seed integer to 1, 2, …, 10. Then, we performed IRAVNet on subsampled BAM files and checked whether IRAVs identified from the original BAM files were detected or not.

**Comparison to machine learning splicing prediction algorithms.** We compared the IRAVNet with existing splicing prediction approaches (MMSplice and SpliceAI) based on machine learning. For each variant at the splice-site regions extracted in the above subsection, we obtained the predictions of splice effects. For MMSplice, we downloaded GTF file from https://ftp.ebi.ac.uk/pub/databases/gencode/Gencode_human/release_39/gencode.v39.annotation.gtf.gz and performed MMSplice version 2.2.0 using predict_all_table function with the settings of "pathogenicity = True, splicing_efficiency = True". We summarized the effect as the maximum across all exons using the max_varEff function when one variant has multiple exons, and the variants whose delta-logit_psi were below −2 were considered to be positive ones. For SpliceAI, we downloaded the pre-calculated annotation (which includes substitutions, 1 base insertions, and 1–4 base deletions within genes) from the Illumina BaseSpace Sequence Hub website, and extracted the score of DS_AL (Delta score (acceptor loss)) or DS_DL (Delta score (donor loss)), and variants whose score were above 0.5 were set to be positive ones. For each approach (IRAVNet, MMSplice, SpliceAI) and for each genomic variant, we divided into positive and negative groups and calculated the Z-value quantified by IRFinder (IRratio and LocalIRratio) as described in the previous subsection.

## Cloud-based platform for detecting intron retention associated variants from Sequence Read Archive

We used our in-house batch job engine on Amazon Web Service, ecsub (https://github.com/aokad/ecsub). This software first launches a virtual machine of an instance type suited to a target task (e.g., transcriptome alignment, gene expression measurement, and so on). Then it performs a series of Extraction Transformation Load (ETL) procedures, in which it extracts input files (e.g. FASTQ files) stored in Amazon S3 to the virtual machines, transforms the input files to output files (e.g. BAM files converted from the FASTQ file by alignment software), and then load the output file to Amazon S3. Finally, ecsub removes the virtual machines used for the ETL procedure. This software basically wraps Amazon Elastic Container Service to implement the above approaches, with additional functions such as selecting the instance type and availability zone of SpotInstance, automatically deleting the master instance using the serverless framework (Amazon Lambda), and so on. This software is inspired by dsub (https://github.com/DataBiosphere/dsub), which is a tool to submit batch jobs on Google Cloud Platform. In this study, the pipeline is composed of four steps; downloading SRA samples, Alignment of transcriptome data, detection of IRAVs using iravnet, and intron retention quantification.

## Investigating generation of premature termination codon and sensitivity to nonsense-mediated-decay

To explore the generation of premature termination codon (PTC) for each IRAV, we chose the longest transcript (defined by RefSeq transcript annotation) harboring the corresponding exon-intron boundary affected by the IRAV. Then, we constructed the transcript with the nucleotide sequence of the retained intron and investigated whether a stop codon occurs before the original termination codon or not. For IRAVs with PTCs, if the PTCs were located before 50 bp upstream of

the last exon–exon junction, then those IRAVs were classified into NMD-sensitive, whereas the remaining PTC harboring IRAVs were deemed as NMD-insensitive.

## Definition and classification of putative pathogenic and drug response intron retention associated variants

We investigated the positional relationships of the IRAVs with pathogenic variants registered in ClinVar VCF file downloaded from ClinVar FTP site (https://ftp.ncbi.nlm.nih.gov/pub/clinvar/vcf_GRCh38/) as of 4th, 2021. The "pathogenic" variant in ClinVar is defined as those whose CLNSIG INFO key is either of "Pathogenic," "Likely_pathogenic," or "Pathogenic/Likely_pathogenic" in this study. For each IRAV, we inspected the positional relationships with pathogenic variants in the ClinVar VCF file. First, when a pathogenic variant with the same genomic position and substitution patterns with the IRAV exist in the ClinVar VCF file, then the IRAVs were classified into Tier1. When a pathogenic splicing variant (MC INFO key is "SO:0001575|splice_donor_variant," or "SO:0001574|splice_acceptor_variant") sharing the same splicing site (3 exonic and 6 intronic bases for splice donor sites, and 6 intronic and 1 exonic base for splice acceptor sites) with the IRAV were observed, the IRAV was designated as Tier2. Next, we searched for a pathogenic truncating variant (MC INFO key is "SO:0001587| nonsense," or "SO:0001589|frameshift_variant") within 30 bp from the corresponding exon–intron boundary at the transcript level, and the pathogenicity rank of the IRAV was set to Tier3 if a variant satisfying the condition was found. Finally, we explored for a pathogenic variant in the same gene with the IRAV, and the IRAV was categorized into Tier4 if one could be found. In the case of Tier4 investigation, we imposed additional constraints, CLNSTAT INFO key is either of "criteria_provided,_multiple_submitters,_no_conflicts," "reviewed_by_expert_panel," "practice_guideline," to focus on variants with solid evidence.

The identification and classification of IRAVs putatively affecting drug response were performed in exactly the same way as above, except that variants whose CLNSIG INFO key is "Drug_response" (instead of "Pathogenic," "Likely_pathogenic," or "Pathogenic/Likely_pathogenic") were used as a reference set for positional relationship comparison with IRAVs.

## Measurement of the amount of intron retention across multiple tissues

We have downloaded the GTEx transcriptome sequence data from SRA and aligned them using STAR, and quantified the amount of intron retention using our in-house program (intron_retention_utils simple_count command) as described in the previous sections. For genotype data, we used exome genotype calls from the GTEx Analysis V7. For each IRAV, we also calculated a $p$-value measuring the difference of the amount of intron retention between samples with and without IRAVs in each tissue via one-sided Wilcoxon rank-sum test using wilcox.test function of R language, and integrated it using Fisher's method across tissues.

## Validation using In vitro assay

The *P3H1* and *SMAD4* mini-gene constructs were generated by inserting the DNA fragments containing the *P3H1* genomic sequence spanning exons 9 and 10 and intervening intron 9, and *SMAD4* genomic sequence from exon 3 to exon 4 in between the BamHI and EcoRI restriction sites of the pcDNA3.0(-) plasmid. Mutagenesis was performed with the primeSTAR Mutagenesis Basal Kit (Takara) with specific primers according to the manufacturer's instructions. For transient transfection experiments, 293T cells were seeded into a 6-well plate one day before transfection of *P3H1* or *SMAD4* minigene constructs in the presence of PEI MAX (polysciences). 36 hours after transfection, cells were collected and RNA was extracted with Favor-Prep Mini Kit (FAVORGEN). Mini-gene-derived and endogenous

transcripts of P3H1 and SMAD4 were analyzed by RT-PCR using specific primers. Primers and oligonucleotides used in RT-PCR reactions were: *P3H1* fwd (mini-gene) CGCAAATGGGCGGTAGGCGTG, *P3H1* fwd (endogenous) AGTCACTGGATGTGAGCAGACTGAC and rev (common) TGAGGGCTTTGAAGACAGTGACAC; SMAD4 fwd (mini-gene) CGCAAATGGGCGGTAGGCGTG, SMAD4 fwd (endogenous) ATACAGAGAACATTGGATGGGAGGCTTC and rev (common) ATTACTCTGCAGTGTTAATC CTGAGAG.

### Reporting summary

Further information on research design is available in the Nature Research Reporting Summary linked to this article.

## Data availability

The list of IRAVs is available through IRAVDB (https://iravdb.io) and Zenodo (https://doi.org/10.5281/zenodo.7045663). Source data are provided with this paper.

## Code availability

The workflow of iravnet is available at GitHub (https://github.com/friend1ws/iravnet).

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

## Acknowledgements

This work is supported by Grant-in-Aid for Scientific Research (KAKENHI 18H03327, 21H03549) and Grand-in-Aid from the Japan Agency for Medical Research and Development (Platform Program for Promotion of Genome Medicine: 20km0405207h9905, Program for an Integrated Database of Clinical and Genomic Information: 20kk0205014h0005, Practical Research Project for Rare/Intractable Diseases: 20ek0109485h0001, Practical Research for Innovative Cancer Control: 21ck0106641h0001), National Cancer Center Research and Development Funds (2020-A-7, 2021-A-3). A.Y. acknowledges support from the Japan Society for the Promotion of Science (JSPS) Home-Returning Researcher Development Research (grant number 19K24691), KAKENHI (grant number 21H04828), the Japan Agency for Medical Research and Development (grant number 21jm0210085h0002), and National Cancer Center Research and Development Funds (2020-A-2). We used the super-computing resource provided by the Human Genome Center (The University of Tokyo) and ROIS National Institute of Genetics. The results shown here are partly based upon data generated by TCGA Research Network (https://cancergenome.nih.gov/) and the Genotype-Tissue Expression (GTEx) Project. The authors want to thank Yuichi Kodama for helpful suggestions on the International Nucleotide Sequence Database Collaboration Policy.

## Author contributions

Y.S. designed the study. Y.S. developed the software for detecting intron retention associated variants. A.O. developed a cloud-based platform for analyzing massive transcriptome sequencing data deposited in Sequence Read Archive and a web-based database. A.O. developed the portal website for IRAVs with assistance from Y.S.; I.O. and H.Y.

performed mini-gene assays under the supervision of A.Y.; K.C., A.K., R.N.M., and N.I. provided computational assistance. Y.S. interpreted and organized the list of intron retention associated variants with assistance from K.K. and A.Y.; Y.S. and A.O. generated figures. Y.S. wrote the manuscript. All authors participated in the discussion and interpretation of the data and results.

## Competing interests

The authors declare no competing interests.
