## [Peer Review File · Nature Communications]

Systematic identification of intron retention associated variants from massive publicly available transcriptome sequencing dataREVIEWER COMMENTS

Reviewer #1 (Remarks to the Author):

This is an interesting and original body of work that makes use of public datasets to identify genetic variants putatively associated with intron retention.

Suggestions

Comparison to existing computational predictions.

There are now quite a few computational methods designed to predict the effects of variants on splicing and I think it is essential to compare how well some of the best / most recent work for identifying intron retention events (when matched DNA-sequence is available). For example some of the following.

Neural networks:

PANGOLIN

MMSplice

SpliceAI

Linear models:

HAL

SMS Score

MaxEntScan

NMD predictions

The authors use the 50nt rule but this only accounts for a minority of NMD-evading stops. There are much better pre-calculated predictions already available such as NMDetective (Lindeboom/Supek Nature Genetics).

Data availability

From a quick scan of the associated website <https://iravdb.io/> I could not find a way to download all of the IRAVs. It would therefore be very difficult for someone to reproduce the analyses or compare the performance of alternative methods to IRAVNet. Please make the data available.

P6 'GT-AT essential splice sites'. Do the authors mean GT-AG ?

abstract. I found this sentence rather grandiose/pompous: 'The new in-silico screening framework proposed here provides a foundation for a platform that can automatically acquire medical knowledge making the most of massively accumulated publicly available sequencing data.'

Reviewer #2 (Remarks to the Author):

In this work the authors present IRAVNet, a method to detect genetic variants that are associated with intron retention from RNASeq data, without the need for WES or WGS. IRAVNet works by scanning RNASeq reads that are in close proximity to splice junctions, finding cases where the read would extend into the adjacent intron and carry a mutation in the positions close to the splice site. It

then compares samples that have the identified mutation (and reads extending into the intron) to samples that do not have the mutation, comparing the fraction of IR in both groups to call intron retention associated variants (IRAVs).

The authors then apply IRAVNet on TCGA data and compare their results to their previous method (Shiraishi et al GR 2018) SAVNet which uses both WES and RNASeq, then apply IRAVNet on the entire SRA RNAseq data, define 4 levels of support for identified IRAV compared to known mutations that are (likely) pathogenic, study specific genes that are enriched in those, test Tier 1-4 cases across GTEX, and perform mini-gene reporter assay to validate two variants.

Overall, there is an impressive amount of work that clearly went into this manuscript (we should also mention above the effort to build a cloud compute infrastructure to process all the data). We also agree with the authors that coming up with clever ways to extract pathogenic variants from the vast SRA, and specifically for RNA splicing which is known to play a role in many disease, is an important endeavor. The authors also spent time analyzing the data, identifying known variants in the literature, and performing two minigene experiments - while having only two does not help much to validate the approach (see below) we do appreciate the effort that went into these experiments.

With that stated, we did have some major concerns that require addressing, as we list below.

1. There is no real assessment of the accuracy of the proposed method:

This is a major problem that goes back to their original 2018 GR paper mentioned above, which suffers from the same problem. Basically, the authors derive some (sensible) filters over junction spanning reads and reads spanning into "introns" (more on this definition below), and continue from there. There is no evaluation using real or synthetic data of FP, FN etc. True, it is hard to do this here where you would need to simulate expression, mutations, coverage etc. But several things are still very doable and should be included to land more credibility to the analysis that follows:

(a)

Test reproducibility: What happens if you compare IRAVNet calls on say 20% of the data vs. another 20% of the data? 30% vs 30%? etc. How many of the variants are being reproduced like that?

(b)

Test that the IR events you are calling are supported by an independent method. This is especially relevant since this was never tested in the GR 2018 paper either. A simple way to do that is use your IRAVNet to classify the samples into two subgroup ("positive" and "negative") and run differential splicing analysis with a method that supports de-novo differential IR detection/quantification. Two methods that come to mind are JUM and MAJIQ. Note that you don't have to do this for ALL variants but for a large enough set across different threshold/expression/#reads to get some confident statistics. True, this would not tell us anything about FN but at least the support by other methods for the events you call as differential IR.

Related to that: The term of IR seems problematic here. At best these are "putative IR". This is because of inherent limitations that are not discussed. Our understanding is that the method here relies on RefSeq as the annotation and junction spanning reads. As such, the reads that go "into an intron" could very well go into an exon extension (alternative 3'/5') that are further out and possibly were not even detected due to low coverage, mapability, etc. Yes, the authors state they look for exon extensions ("we removed those completely included in the exonic regions of other genes or isoforms") but since they use RefSeq as the base they are clearly missing many other isoforms of the same gene. Also, unlike the IR used by the methods mentioned above or Braunschweig et al GR 2014, the method used here (and in the GR 2018 as well) does not test for any coverage along the intron. It would be interesting to see how many of the called IR in this method are indeed supported by these other methods, and how many are supported but as other types of AS.

(c) You could evaluate IR performance by creating simulated data form say GTEX with an RNAseq

simulator over say GENCODE annotation and whatever GE estimates are available for those transcripts, then quantified the cases where there is IR and test the accuracy of your IR method to detect those. Yes, we are aware this is a considerable undertaking and the focus here is not on a comprehensive IR detection method, so this is probably not a must in this case. Still we feel it is worth pointing out that some synthetic data evaluation is possible and previous publications on splicing analysis methods used such evaluations.

2.

Unclear relevance of IRaVs detected:

(a) Much of the data in SRA comes from cell lines. The authors filter against common variants. It's not clear how many of the IRaVs reported are unique to cell lines and never occur in the human population/samples.

(b) The analysis seems to lack any annotation of the SRA samples that would help with cell line specific mutations mentioned above but also to detect tissue/cell specific effects, possibly increasing substantially the amount of variants and making them more "interesting" (tissue/condition specific). As it stands, this huge analysis effort ends up with ~3K variants total, of whom unknown fraction is driven by cell lines.

(c) The GTEx analysis seems to give very few hits and they don't look so good. Yes, Fig6a looks nice, but these are 3 and the supplementary figures have several more and different tier, not looking so great. Related to that: To the variants effect on GTEx cluster by tissue type?

Other comments/concerns:

The authors repeatedly use the term "cause" which is clearly not the case (except of the 2 validated by minigene). Even the name "IRaV" uses the term "associated". True, by focusing on variants very close to the splice site etc. there is a much higher chance these are indeed causal, but the analysis itself is based on statistical association.

It is not clear how many variants are thrown out at the various steps of the filtering process: gnomAD allele frequency 0.01, MaxEntScore change > 2.5, etc.

How does the method handle batch effects? My guess is that there is no handling of that even in data where such information exists. That should be discussed. If anything, maybe use the detection of an IRaV across multiple datasets as a "proof" that these are not batch/dataset specific?

The authors use the term "validation" loosely. The only actual validation is the minigene reporter assays. Figure 2 is by no means 'validation' of any sort. It is, at best, assessing the results of IRaVNet compared to a similar IR calling pipeline but when exome sequencing data is available.

Related to that:

The claim of FP of 0.93% about the method is highly misleading. If we understood correctly, the only thing this test did was to assess if the false positive of mutations compared to those detected by exome sequencing. So this just means that the pipeline identifies "real" mutations. Furthermore, if we understood the subsequent analysis correctly compared to SAVNet then, when using the same IR detection/quantification method, IRaVNet has Sensitivity of ~43.7% for somatic mutations and FDR of ~26%: The authors report 348/1321 variants not detected by SAVNet as "novel" - but if the same IR method uses more complete genetic variants calling and does not find them aren't these suspected FP and not "novel"?

Fig 2c: This was unclear. The z-score here seems like a circular argument because the method looks for that enrichment, no? If these are based on WES mutations calls, then it goes back to our question about estimating FP above.

Fig3a: Unclear what is shown there.

Fig4b: How do we know these plots are not driven by FP/noise? Related to comments above about accuracy and reproducibility.

In summary, there has clearly been a tremendous amount of effort, thinking, and analysis that went into this work. We very much appreciate that. But at the same time, the end result is not as comprehensive as one would hope. While the filters probably help make the results substantially less noisy, there is a price for focusing on very few positions around splice sites in RefSeq, and with no tissue/condition information integrated. The accuracy of the method is still unknown. As the authors note, there is also a substantial price to repeat this pipeline if one would ever want to extend the list of IRAV with additional data. These things together dampens the overall impact of the work and our enthusiasm, even though the topic is important and the scope of work is impressive.

Reviewer #3 (Remarks to the Author):

The authors present IRAVNet, a method that detects and provides a tier-based ranking system for splicing variants (specifically intron retention associated variants (IRAVs)) linked to disease variants. This method negates previous requirements for linked transcriptomic and genomic data, and can therefore be run on large numbers of publicly available transcriptomic datasets. This approach allows the identification of far more significant splicing variants than previous methods. The authors provide a website to explore identified splicing variants as well as their general pipeline for running IRAVNet on publicly available SRA data.

The approach, based on base mismatches near the intron-exon boundary, is clear, reasonable, and reproducible, and the attached resources are well-made and easy to use. In particular, variants identified as Tier 1 can provide a valuable resource as a validation set of disease-related splicing variants. In terms of functional interpretation of the identified splicing variants, the mini-gene approach across two events is reasonable. Additionally, using ClinVar to overlap the set of variants identified is a reasonable validation approach.

Minor critiques:

- While I understand the scope of the tool is to find high-quality variants, the exclusion of variants on only one haplotype seems to be a limitation. I suspect there are likely some events that show strong effects that happen to be somatic or only on one strand. Perhaps the authors can comment on how many events are lost at each stage on an example data set and how many false-positives/true-positives they think exist due to this filtering heuristic.
- On the website: it would be useful to have a centralized list of all variants across all genes grouped by tier, especially a list of all Tier 1 variants and their associated gene.
- While the tools used for the quantification of intron retention are open source (https://github.com/friend1ws/intron_retention_utils), they do not appear to be peer reviewed. Why was this approach used as opposed to one of the established methods such as MISO or SplAdder?

Response to Reviewers' comments

Reviewer 1

This is an interesting and original body of work that makes use of public datasets to identify genetic variants putatively associated with intron retention.

We thank the Reviewer for the kind compliments.

Suggestions

1) Comparison to existing computational predictions.

There are now quite a few computational methods designed to predict the effects of variants on splicing and I think it is essential to compare how well some of the best / most recent work for identifying intron retention events (when matched DNA-sequence is available). For example some of the following. Neural networks: PANGOLIN, MMSplice, SpliceAI. Linear models: HAL, SMS Score. MaxEntScan.

We completely agreed with the Reviewer that the comparison between IRAVNet and machine-learning based prediction methods to predict the splicing variant effects would give us helpful insight. We performed a comparison using 1000 genomes whole-genome sequencing data where matched transcriptome is available. Please see Supplementary Figure 6b in the newly added "Application on GEUVADIS dataset" subsection in the Result section. Overall, the effects of variants identified by IRAVs were far more substantial than those predicted by MMSplice or SpliceAI. We see two reasons for this. One is that IRAVNet directly investigates sequence data if there is any variant and has an advantage over a purely predictive approach from the nucleotide sequences. The other is that, although current machine learning-based approaches may predict whether there are some "splicing aberrations," it is still difficult to distinguish the detailed consequences such as exon skipping, alternative 5' or 3'SS, and intron retention. We added the following description to the new section of "Application on GEUVADIS dataset:"

Next, we compared our approach with machine learning-based splicing effect prediction approaches (SpliceAI³, MMSplice¹⁸). The variants identified by IRAV showed more substantial enrichment of intron retention events than those predicted by machine learning prediction methods (Supplementary Figure 6b). We believe this is because IRAVNet approach directly observes the variants from transcriptome sequencing data and may have an advantage over purely predictive approaches based on the nucleotide sequences. In addition, splicing aberrations caused by genomic variants are heterogeneous (including exon skipping, alternative 5 and 3' splice site), and current machine learning approaches have not been specifically trained to predict intron retention.

2) NMD predictions

The authors use the 50nt rule but this only accounts for a minority of NMD-evading stops. There are much better pre-calculated predictions already available such as NMDetective (Lindeboom/Supek Nature Genetics).

Thank you very much for pointing this out. Although their study (Lindeboom et al., Nature Genetics, 2019) generated excellent resources for NMD prediction, they focused only on nonsense mutations and frameshift indels, assuming that the backbone transcripts were invariant. Since the forms of transcripts may change by splicing mutations, their prediction data cannot be used directly to splicing mutations. Therefore, we have only added their paper to the reference list. However, as we

discussed in the Discussion section, we agree with the Reviewer that using the 50nt rule alone is not sufficient, and we would like to incorporate the outcome of future studies related to NMD predictions.

3) Data availability

From a quick scan of the associated website <https://iravdb.io/> I could not find a way to download all of the IRAVs. It would therefore be very difficult for someone to reproduce the analyses or compare the performance of alternative methods to IRAVNet. Please make the data available.

We made all the IRAVs downloadable from <https://iravdb.io/download>.

4) P6 'GT-AT essential splice sites'. Do the authors mean GT-AG ?

We apologize to the Reviewer for this confusion. We corrected the description.

5) abstract. I found this sentence rather grandiose/pompous: 'The new in-silico screening framework proposed here provides a foundation for a platform that can automatically acquire medical knowledge making the most of massively accumulated publicly available sequencing data.'

We thank the Reviewer for this comment. We have weakened the wording according to the recommendation. Still, we believe that we can automate the detection of various types of pathogenic variants by devising similar approaches to IRAVNet and applying them to the Sequence Read Archive data that will increasingly accumulate sequencing data. We are now developing a novel method for detecting variants creating splice sites that generate new splicing junctions. This paper would be a basis for a series of studies that will follow.

Reviewer 2

In this work the authors present IRAVNet, a method to detect genetic variants that are associated with intron retention from RNASeq data, without the need for WES or WGS. IRAVNet works by scanning RNASeq reads that are in close proximity to splice junctions, finding cases where the read would extend into the adjacent intron and carry a mutation in the positions close to the splice site. It then compares samples that have the identified mutation (and reads extending into the intron) to samples that do not have the mutation, comparing the fraction of IR in both groups to call intron retention associated variants (IRAVs).

The authors then apply IRAVNet on TCGA data and compare their results to their previous method (Shiraishi et al GR 2018) SAVNet which uses both WES and RNASeq, then apply IRAVNet on the entire SRA RNAseq data, define 4 levels of support for identified IRAV compared to known mutations that are (likely) pathogenic, study specific genes that are enriched in those, test Tier 1-4 cases across GTEx, and perform mini-gene reporter assay to validate two variants.

Overall, there is an impressive amount of work that clearly went into this manuscript (we should also mention above the effort to build a cloud compute infrastructure to process all the data). We also agree with the authors that coming up with clever ways to extract

pathogenic variants from the vast SRA, and specifically for RNA splicing which is known to play a role in many disease, is an important endeavor. The authors also spent time analyzing the data, identifying known variants in the literature, and performing two minigene experiments - while having only two does not help much to validate the approach (see below) we do appreciate the effort that went into these experiments.

We thank the Reviewer's kind compliments that "there is an impressive amount of work that clearly went into this manuscript," "we should also mention above the effort to build a cloud compute infrastructure to process all the data," and "coming up with clever ways to extract pathogenic variants from the vast SRA, and specifically for RNA splicing which is known to play a role in many diseases, is an important endeavor."

With that stated, we did have some major concerns that require addressing, as we list below.

1) 1. There is no real assessment of the accuracy of the proposed method:

This is a major problem that goes back to their original 2018 GR paper mentioned above, which suffers from the same problem. Basically, the authors derive some (sensible) filters over junction spanning reads and reads spanning into "introns" (more on this definition below), and continue from there. There is no evaluation using real or synthetic data of FP, FN etc. True, it is hard to do this here where you would need to simulate expression, mutations, coverage etc. But several things are still very doable and should be included to land more credibility to the analysis that follows:

- a. Test reproducibility: What happens if you compare IRAVNet calls on say 20% of the data vs. another 20% of the data? 30% vs 30%? etc. How many of the variants are being reproduced like that?**

We thank the Reviewer for this valuable suggestion. We subsampled the sequence reads and checked how much ratio of IRAVs could be recaptured for several subsample ratios (0.1 increments from 0.1 to 0.9) for testing the reproducibility. Whether an IRAV could be identified for a subsampled set depends on the expression levels. However, generally, a considerable amount of IRAVs could be reproduced (Supplementary Figure 6a). We added the following statement to the new section "Application on GEUVADIS dataset:"

We also applied IRAVNet to downsampled transcriptome sequencing data to assess the robustness of IRAVNet approach, confirming a considerable amount of IRAVs could be reproduced (Supplementary Figure 6a).

- b. Test that the IR events you are calling are supported by an independent method. This is especially relevant since this was never tested in the GR 2018 paper either. A simple way to do that is use your IRAVNet to classify the samples into two subgroup ("positive" and "negative") and run differential splicing analysis with a method that supports de-novo differential IR detection/quantification. Two methods that come to mind are JUM and MAJIQ. Note that you don't have to do this for ALL variants but for a large enough set across different threshold/expression/#reads to get some confident statistics. True, this would not tell us anything about FN but at least the support by other methods for the events you call as differential IR.**

Related to that: The term of IR seems problematic here. At best these are "putative IR". This is because of inherent limitations that are not discussed. Our understanding is that the method here relies on RefSeq as the annotation and junction spanning reads. As such, the reads that go "into an intron" could very well go into an exon extension (alternative 3'/5') that are further out and possibly were not even detected due to low coverage, mapability, etc. Yes, the authors state they look for exon

extensions (“we removed those completely included in the exonic regions of other genes or isoforms”) but since they use RefSeq as the base they are clearly missing many other isoforms of the same gene. Also, unlike the IR used by the methods mentioned above or Braunschweig et al GR 2014, the method used here (and in the GR 2018 as well) does not test for any coverage along the intron. It would be interesting to see how many of the called IR in this method are indeed supported by these other methods, and how many are supported but as other types of AS.

We performed several experiments and added a new section “Application on GEUVADIS dataset.” We tested whether the IR events are also supported by the different methods from our in-house approach. We adopted MAJIQ and IRFinder, which seem to be relatively well maintained (the approach used in Braunschweig et al. GR 2014 has not been implemented, and we could not use it, unfortunately). Using IRFinder, we derived two indices, LocalIRRatio (measuring the intron retention ratio around the splice-sites) and IRRatio (measuring the amount of sequence coverage along the intron). When we divided into positive and negative groups, most of IRAVs (97.0%) showed a significant difference between the groups based on LocalIRRatio via IRFinder, reproducing the result in Figure 2c. When measuring via intron-wide sequencing depth (IRRatio via IRFinder, MAJIQ), the effect became slightly weaker, possibly because the consequences of splicing causing variants and their appearance in transcriptome data are often complex. We add a new subsection “Application on GEUVADIS dataset” and described the result above. Please also see Supplementary Figure 6b, which evaluates IRAVNet through the comparison with machine-learning based prediction approaches.

To explore whether these IRAVs have certain effects on intron retention, we divided samples into IRAV positive and negative groups based on genotyping via whole-genome sequencing data and compared the amount of intron retention measured by several methods (IRFinder¹⁶ and MAJIQ¹⁷) for each IRAV. When we measured the local intron retention ratio around the exon-intron boundary (IRFinder LocalIRRatio), the effect on intron retention was rather clear for most IRAVs, corroborating that these IRAVs were likely to induce intron retention (Z-value ≥ 2 and p-value ≤ 0.01 for 97.0% IRAVs, see Supplementary Figure 4 and 5). When measuring via intron-wide sequencing depth (IRFinder IRRatio, MAJIQ), the effect becomes slightly weaker, possibly because the consequences of splicing causing variants and their appearance in transcriptome data are often complex. We also applied IRAVNet to downsampled transcriptome sequencing data to assess the robustness of IRAVNet approach, confirming a considerable amount of IRAVs could be reproduced (Supplementary Figure 6a).

We also thank the reviewer for pointing out the possibility that intron retention predicted via IRAVNet is based on unannotated transcripts by RefSeq. We checked whether the splice-sites targeted by IRAVs were contained by the exons by GENCODE basic gene annotation, and out of 23,849 IRAVs (2.2%), 517 IRAV splice-sites were actually contained by any of GENCODE annotation. We do not believe that most of these 517 IRAVs are false positives because of the following reasons:

- (1) We have pre-filtered regions where we can observe common intron retention using 742 control (non-cancer) data from TCGA. Therefore, these IRAV-containing transcripts are not likely ones that are frequently observed.
- (2) These IRAVs included seven variants registered as Pathogenic or Likely_pathogenic in ClinVar as splice-site variants (Tier1 putative pathogenic IRAVs). Also, several IRAVs were also identified by the previous approach (SAVNet, Shiraishi et al., Genome Research, 2018) that is based on a statistical association.

Nevertheless, we have chosen to remove them in this paper to make the list of IRAVs more reliable. The various statistics have changed slightly accordingly. However, most key IRAVs mentioned in the manuscript, such as those observed in cancer driver genes, remain unchanged.

- c. **You could evaluate IR performance by creating simulated data from say GTEX with an RNAseq simulator over say GENCODE annotation and whatever GE estimates are available for those transcripts, then quantified the cases where there is IR and test the accuracy of your IR method to detect those. Yes, we are aware this is a considerable undertaking and the focus here is not on a comprehensive IR detection method, so this is probably not a must in this case. Still we feel it is worth pointing out that some synthetic data evaluation is possible and previous publications on splicing analysis methods used such evaluations.**

IRAVNet detects IRAVs by detecting the variant around the splice-site from the transcriptome sequence data and confirms that the variant is supported specifically by intron-retention reads. Therefore, in order to generate synthetic data, we need to consider not simply gene expression but also other factors such as sequencing errors, alignment errors, “natural” intron retention events caused without mutations, and complicated splicing changes due to mutations taking into account co-occurrence of other types of splicing changes. Based on these reasons, we think it is unfortunately difficult to generate synthetic data that naturally mimics transcriptome data in reality.

2) Unclear relevance of IRAVs detected:

- a. **Much of the data in SRA comes from cell lines. The authors filter against common variants. It's not clear how many of the IRAVs reported are unique to cell lines and never occur in the human population/samples.**
- b. **The analysis seems to lack any annotation of the SRA samples that would help with cell line specific mutations mentioned above but also to detect tissue/cell specific effects, possibly increasing substantially the amount of variants and making them more “interesting” (tissue/condition specific). As it stands, this huge analysis effort ends up with ~3K variants total, of whom unknown fraction is driven by cell lines.**

As the Reviewer pointed out, much of the transcriptome data in SRA came from cell lines.

We definitely agree with the Reviewer that it is important to see whether a specific variant identified in SRA is unique to cell lines or not and to clarify the association between each IRAV and disease. We randomly chose 80 putative pathogenic IRAVs (20 IRAVs per Tier) and investigated the source by manually checking the abstract of the Sequence Read Archive portal site, which revealed that at least 54 out of them were identified from human samples (Tier1: 15 / 20, Tier2: 15 / 20, Tier3: 9 / 20, Tier4: 15 / 20).

The metadata of samples registered in SRA is currently not well organized at all and it is extraordinarily challenging to systematically annotate the tissue type or to determine whether a data sample is derived from a cell line or not. A computationally automated annotation pipeline for text mining would be necessary to achieve this goal, which we think would be beyond the current study. We were fully aware of this limitation at the first submission, and we mentioned this point in the Discussion session as follows:

The Sequence Read Archive, which we mainly used in this paper, is not well organized in terms of metadata, making it difficult to perform association studies with some phenotypes. It may be helpful to devise and apply computational prediction methods for basic information such as race, tissue, and cancer status.

c. The GTEx analysis seems to give very few hits and they don't look so good. Yes, Fig6a looks nice, but these are 3 and the supplementary figures have several more and different tier, not looking so great. Related to that: To the variants effect on GTEx cluster by tissue type?

As an objective criterion, we provided a p-value for each IRAV by measuring the difference of the intron retention ratios between samples with and without IRAVs via Wilcoxon rank sum test for each tissue and integration the p-values across tissues by Fisher's method for each IRAV in Figure 6a and Supplementary Figure 13-14, and add the description to each figure legend as follows:

(Figure 6a) The p-values measuring the differences of the intron retention ratios between samples with and without IRAVs via Wilcoxon rank sum test for each tissue and integration by Fisher's method were 1.82×10^{-42} , 2.91×10^{-7} , and 1.22×10^{-69} , respectively from the left panel to the right.

The p-values were 3.74×10^{-4} at maximum and mostly below 1.0×10^{-8} in these figures, indicating significant effects on intron retention when viewing tissue-wide. We completely agree with the Reviewer that whether the effects of intron retention have some tendency to cluster to tissue types is a fundamental question. However, since our study has been focused on rather rare variants (≤ 0.01 allele frequencies via gnomAD) and there were not enough IRAVs detected in about 500 GTEx samples it is difficult to verify these effects, unfortunately.

3) The authors repeatedly use the term "cause" which is clearly not the case (except of the 2 validated by minigene). Even the name "IRAV" uses the term "associated". True, by focusing on variants very close to the splice site etc. there is a much higher chance these are indeed causal, but the analysis itself is based on statistical association.

We thank the Reviewer for this query. As the Reviewer pointed out, most IRAVs identified in this paper have not been validated to truly "cause" intron retention. On the other hand, we would like to mention that our approach is not based on statistical associations. Our method is based on the phenomenon that, when variants cause intron retention, those are always found in reads that support intron retentions and confirm these facts within the detection pipeline (Please see Supplementary Figure 1).

4) It is not clear how many variants are thrown out at the various steps of the filtering process: gnomAD allele frequency 0.01, MaxEntScore change > 2.5, etc.

Filtering by the allele frequency (AF) via gnomAD filtering had been performed at a very early stage. However, if neither filtering by gnomAD AF nor MaxEntScore were performed, 50,446 IRAVs would remain (before removing non-coding or NMD nonsensitive variants). Then, removing the variant of AF > 0.01 via gnomAD, 47,356 IRAVs would remain.

5) How does the method handle batch effects? My guess is that there is no handling of that even in data where such information exists. That should be discussed. If anything, maybe use the detection of an IRAV across multiple datasets as a "proof" that these are not batch/dataset specific?

First, please note that IRAVNet can be performed for a single transcriptome data. Even for the organized dataset such as TCGA, IRAVNet was performed for each transcriptome sequencing data separately. This is clear merit compared to, e.g., our previous methods (SAVNet), which need to specify the group of input data. Therefore, it is not necessary to consider batch effects when using IRAVNet. We added this point to the Introduction in the manuscript as follows:

Another advantage of IRAVNet is that, unlike previous methods that rely on statistical associations, this pipeline can be run on single transcriptome sequencing data without gathering or specifying a set of data, making a systematic analysis of public repositories much easier.

6) The authors use the term “validation” loosely. The only actual validation is the minigene reporter assays. Figure 2 is by no means ‘validation’ of any sort. It is, at best, assessing the results of IRAVNet compared to a similar IR calling pipeline but when exome sequencing data is available. Related to that: The claim of FP of 0.93% about the method is highly misleading. If we understood correctly, the only thing this test did was to assess if the false positive of mutations compared to those detected by exome sequencing. So this just means that the pipeline identifies “real” mutations. Furthermore, if we understood the subsequent analysis correctly compared to SAVNet then, when using the same IR detection/quantification method, IRAVNet has Sensitivity of ~43.7% for somatic mutations and FDR of ~26%: The authors report 348/1321 variants not detected by SAVNet as “novel” - but if the same IR method uses more complete genetic variants calling and does not find them aren’t these suspected FP and not “novel”?

Thank you very much for pointing this out. We modified the term of validation to “assessment” for the evaluation using TCGA data.

As the reviewer suggested, 0.93% (now changed to 0.96% because of the filtering of by GENCODE annotation) was the false-positive ratio in terms of mutation status at the level of the genome. At this stage, there is a possibility that the IRAV is just the genomic variant not causing splicing changes. Therefore, we modified the expression so that this point would be clear as follows:

This revealed that the ratio of false positives in terms of genomic mutation status was estimated to be as low as 0.96% (Figure 2b).

Please note the following breakdown of the variant newly identified by IRAVNet, along with descriptions on the characteristics of SAVNet:

1. SAVNet generates the list of pairs of genomic variants and associated splicing changes (exon skipping, alternative 5' (3') splice site, intron retention). One variant can be associated with multiple splicing changes. Here, among the 331 (which is the statistics at the first submission) IRAVs which were not detected by SAVNet, 118 were in fact identified as SAVs (splicing associated variants) that were associated with other splicing changes than intron retention. Since one variant can cause multiple splicing changes, it is possible that these 118 IRAVs were associated with at least some kind of intron retention, probably be missed by the criteria of SAVNet.
2. SAVNet requires a list of pre-called somatic mutations as well as RNA-seq data. Since the amount of sequence depth of exome sequence data in TCGA is especially large, somatic mutation calling is a subtle procedure that can be affected algorithmic choice and post-filtering. Among the remaining 213 IRAVs not identified by SAVNet, 76 were not in the somatic mutation list used by SAVNet, in which a number of stringent filtering were performed (see Shiraishi et al., Genome Research, Method section). In the validation of the mutation status of each IRAV in Figure 2a, we adopted rather simple criteria because the variant is already observed in RNA-seq, and the prior probability that this IRAV is a genuine genomic variant is high if there is a variant supporting read in exome data.
3. SAVNet, in a very simple explanation, performs statistical tests of splicing changes between samples with and without neighboring mutations. However, since there can be multiple mutations (e.g., mutations at the same exon-intron boundaries) that can cause the same types of splicing changes or one mutation can cause many types of splicing changes, we developed a “network” model. Although the network model of SAVNet performed well in most cases, it may fail to identify plausible variants in some cases. Among the remaining

variants, 15 were variants in *TP53*, and we were able to identify other somatic driver genes in the associated cancer types (such as *MEN1* in ACC, *SMAD4* in COAD, *PIK3R1* in UCEC). Based on the observations above, a certain number of variants that were not detected in the previous study (using SAVNet) and identified in the current study (using IRANet) are, if not all, plausible ones. At the same time, we agree with the Reviewer that the term “novel” was too simplistic to describe the above situation, and we have modified the statement as follows: *IRANet identified 331 “new” variants, in the sense that they were not identified by SAVNet or identified as associated with other types of splicing changes than intron retention (Supplementary Figure 2).*

7) Fig 2c: This was unclear. The z-score here seems like a circular argument because the method looks for that enrichment, no? If these are based on WES mutations calls, then it goes back to our question about estimating FP above.

Please note that IRANet does not use any statistical association or enrichment information of intron retention. IRANet identifies variants directly observing the mismatch in the transcriptome alignment and confirms that it is specific to the reads that support the corresponding to intron retention event. The Z-score of the enrichment of intron retention for each cohort group can be one measure for cross-checking.

8) Fig3a: Unclear what is shown there.

We apologize for the insufficient explanation for Figure 3a. We added explanations of a specific example to clarify the content as follows:

For example, in the rows [1.0, 2.0), Zero, one, two, three, and four or more IRANets were identified in 51,793, 6,600, 1,187, 419, and 659 transcriptome sequence data, respectively, whose base count are equal to or more than 1.0 Gbp and less than 2.0 Gbp.

9) Fig4b: How do we know these plots are not driven by FP/noise? Related to comments above about accuracy and reproducibility.

We have already discussed the accuracy and reproducibility of IRANet and other tools above.

Reviewer 3

The authors present IRANet, a method that detects and provides a tier-based ranking system for splicing variants (specifically intron retention associated variants (IRANets)) linked to disease variants. This method negates previous requirements for linked transcriptomic and genomic data, and can therefore be run on large numbers of publicly available transcriptomic datasets. This approach allows the identification of far more significant splicing variants than previous methods. The authors provide a website to explore identified splicing variants as well as their general pipeline for running IRANet on publicly available SRA data.

The approach, based on base mismatches near the intron-exon boundary, is clear, reasonable, and reproducible, and the attached resources are well-made and easy to use. In particular, variants identified as Tier 1 can provide a valuable resource as a validation set of disease-related splicing variants. In terms of functional interpretation of the identified splicing variants, the mini-gene approach across two events is reasonable. Additionally, using ClinVar to overlap the set of variants identified is a reasonable validation approach.

We thank the Reviewer for the kind compliments that “The approach, based on base mismatches near the intron-exon boundary, is clear, reasonable, and reproducible”, and “the attached resources are well-made and easy to use”.

Minor critiques:

1) While I understand the scope of the tool is to find high-quality variants, the exclusion of variants on only one haplotype seems to be a limitation. I suspect there are likely some events that show strong effects that happen to be somatic or only on one strand. Perhaps the authors can comment on how many events are lost at each stage on an example data set an how many false-positives/true-positives they think exist due to this filtering heuristic.

Thank you very much for the suggestion. In the IRAVNet approach, we identified mismatch bases around the splice-site in the transcriptome sequence alignment and confirmed whether this is specific to intron-retention supporting reads (Please see Supplementary Figure 1). Therefore, this methodology detects events in which intron retention is caused by mutations in a splice site where intron retention does not normally occur.

However, it may not be sufficient to detect mutations that cause a quantitative change, such as a mutation that results in stronger intron retention at the innately intron retention-intolerant splice site, where there are already some intron retentions. To identify these variants, other approaches relying on statistical associations would be necessary. Although we were able to detect many disease-associated mutations even at the current stage, future methodological improvements could further increase our detection power. We included the following description in the Discussion:

There are several caveats in the IRAVNet approach. We identified mismatch bases around the splice-site in the transcriptome sequence alignment and confirmed whether this is specific to intron-retention supporting reads. Therefore, this methodology detects events in which mutations cause intron retention in a splice site where intron retention does not commonly occur. However, it may not be sufficient to detect mutations that cause a quantitative change, such as a mutation that results in stronger intron retention at the innately intron retention-intolerant splice site, where there are already some intron retentions.

2) On the website: it would be useful to have a centralized list of all variants across all genes grouped by tier, especially a list of all Tier 1 variants and their associated gene.

Following the Reviewer’s recommendation (it was also requested by Reviewer 1), we made all the IRAVs downloadable from <https://iravdb.io/download>.

3) While the tools used for the quantification of intron retention are open source (https://github.com/friend1ws/intron_retention_utils), they do not appear to be peer reviewed. Why was this approach used as opposed to one of the established methods such as MISO or SplAdder?

We thank the Reviewer for this comment. We have used the tool “intron_retention_utils” in our previous research and it was peer-reviewed (Shiraishi et al., Genome Research, 2018). Although we used this tool as we were very familiar with it, we also added the evaluation of intron retention using other established tools such as IRFinder and MAJIQ. See the newly added Result section “Application on GENVADIS dataset.”

REVIEWER COMMENTS

Reviewer #1 (Remarks to the Author):

The authors have satisfactorily addressed my concerns.

Reviewer #2 (Remarks to the Author):

The authors have done a substantial amount of work to address the three reviewers' many comments and concerns. Some key issues still remain.

First, before going into details regarding the additions made, a more general comment: We realized after reading all the reviewers comments and responses that what we, the reviewers, should have pointed out and asked for direct comparison not necessarily/only ML based predictions (Reviewer #1) or splicing quantification methods (Reviewer #2) but more inline with Reviewer #3 comments - existing pipelines to (a) call variants from RNA-Seq (e.g. the highly used GTAK) (b) perform sQTL analysis on those. These two components, in a nutshell, is what the IRAVNet method proposed here is doing and is most similar to. To our view, IRAVNet makes a conscious decision to perform variants of (a) and (b) above on a very specific set: Only call variants close to a splice site, and only consider cases where intron reads support the variant but not the reference allele. By making such a design choice IRAVNet significantly limits the search space and the ability to find variants causing splicing changes, but importantly (1) makes efficient processing of the entire SRA for such analysis feasible (2) increases the chance (i.e. reduce FDR) that variants called are indeed casual for IR. It is unreasonable, in the mind of this reviewer, to now come back to the authors and require a whole new line of comparisons. But, at the very least, the authors should include in the paper a very clear and detailed discussion about the connections to procedures for (a)+(b) as discussed above.

Now for specific comments:

(1)

On page 2 the authors state:

"unlike previous methods that rely on statistical associations". This is along the lines of their response as well but we strongly disagree with this assertion. IRAVNet is based on statistical association and the claims it's not seem to reflect false logic. Yes, IRAVNet can work on a single transcriptome but that's just because of the way it is defined to filter events. At the end of the day, it is looking for IR reads with variants and the lack of reads without the variants. Seeing enough of those, it calls it an IRAV. Is that not a statistical association?? There is definitely not a probabilistic generative model, nor causal modeling, so I'm not sure why the authors make such claims and these should be removed. This view is also reflected in the authors response to my comments which reads: "...we would like to mention that our approach is not based on statistical associations. Our method is based on the phenomenon that, when variants cause intron retention, those are always found in reads that support intron retention and confirm these facts within the detection pipeline" - again, this argument is inherently flawed. What we have here is an association between detecting IR reads and having these specific variants in those IR reads.

Soon after that on the same page the authors write:

"We confirmed that intron retention associated variants (IRAVs) can be detected with high sensitivity and accuracy".

Actually, the results presented do not support that statement and the basic logic of the method does not support it either. Sensitivity is the amount of TP from those defines as positive. There is no estimate of the total amount of positive (i.e. variants that cause IR) anywhere in the paper. Even the authors themselves seem aware of that. On the following page 3 they state: "Therefore, IRAVNet

achieves a certain level of sensitivity and a high rate of precision". We would agree that the results in the paper, where they compare to SAVNet and IRFinder/MAJIQ, support high precision or low FDR.

(2)

The comparison to IRFinder/MAJIQ is unsatisfactory. The whole point of using such methods is that you could get an estimate of the changes in PSI (IR in this case) in the group with the variant. Instead, the authors choose to lose all the information and just report a z-score between the two groups. Given that there are only IR reads with the variant it is not surprising you would see a significant difference in z-score between the two groups - that is more of a sanity check (an important one!) than any insight into what is being found/reported. For example, running MAJIQ-HET the authors could get a distribution of PSI values in each group with matching statistics. Also, are the p-values for permutations taking into account multiple testing?

(3)

The comparison to ML based predictions is unsatisfactory and unclear. There is one line in the caption which explains the analysis. Nothing is defined there. What are the FALSE and TRUE groups here? Are the same variants considered? If you take the same variants detected by IRVNET proximal to splice sites and ran them via these algorithms and compared the correlation of their scores to observed IR or cases where IRFinder/MAJIQ report significant IR, are these predictions so much different/less accurate than the quantification from the RNASeq reads for these variants with IRVNET? Basically the question is if you used the variant calling from RNASeq for these limited regions and just ran ML prediction algorithms on those, would you get a different result? The plots are also unclear (what is the y-axis? Can't see anything about the concentration of points or stats about the distribution in current plots)

(4)

Page 5: "Most of the IRAVs associated with coding regions were predicted to create premature termination codons" - How? This is not defined. 50nt rule from where? How do you know which isoform? The explanatory comment that follows "(premature termination codons are located before the 50bp upstream of the last exon-exon junction)" is not an accurate statement by itself. Clearly there are cases that do not follow this as the authors are well aware. They probably meant to say that they use this as a rule to call PTC (but from which isoform?) - please fix and clarify.

(5)

Page 7, section on drug response is unclear and underdeveloped. At the very least, define what is the input data, what metrics are used for drug responses etc.

(6)

My question about reproducibility was not about subsampling the data (power) but what happens if you only had say 25%, or 50% of the samples and you applied the same procedure on two subsets of 25% of the data and compared the results - how many of the IRVNET would have been reproduced?

Reviewer #3 (Remarks to the Author):

The authors have sufficiently responded to my critique. Nice work. Thank you.

Response to Reviewers' comments

Reviewer 1

The authors have satisfactorily addressed my concerns.

Thank you very much for your important suggestions and comments.

Reviewer 2

The authors have done a substantial amount of work to address the three reviewers' many comments and concerns. Some key issues still remain.

Thank you very much for acknowledging that "The authors have done a substantial amount of work to address the three reviewers' many comments and concerns."

First, before going into details regarding the additions made, a more general comment: We realized after reading all the reviewers comments and responses that what we, the reviewers, should have pointed out and asked for direct comparison not necessarily/only ML based predictions (Reviewer #1) or splicing quantification methods (Reviewer #2) but more inline with Reviewer #3 comments - existing pipelines to (a) call variants from RNA-Seq (e.g. the highly used GTAK) (b) perform sQTL analysis on those. These two components, in a nutshell, is what the IRAVNet method proposed here is doing and is most similar to. To our view, IRAVNet makes a conscious decision to perform variants of (a) and (b) above on a very specific set: Only call variants close to a splice site, and only consider cases where intron reads support the variant but not the reference allele. By making such a design choice IRAVNet significantly limits the search space and the ability to find variants causing splicing changes, but importantly (1) makes efficient processing of the entire SRA for such analysis feasible (2) increases the chance (i.e. reduce FDR) that variants called are indeed casual for IR. It is unreasonable, in the mind of this reviewer, to now come back to the authors and require a whole new line of comparisons. But, at the very least, the authors should include in the paper a very clear and detailed discussion about the connections to procedures for (a)+(b) as discussed above.

Thank you very much for excellently summarizing our approach. In response to the reviewer's comment, we added the following statement in the Discussion section that attempts to discuss the relationship with sQTLs as follows:

Previous approaches such as splicing QTL^{14,47} evaluate the association between genomic variant status and the amount of splicing changes across samples. However, IRAVNet can be interpreted as evaluating the association differently: It examines the co-occurrence of mutations and splicing events (intron retention) across short reads. This can be achieved by drawing out the multi-layered information provided by the sequencing data (in this case, the presence or absence of both splicing changes and genetic mutations can be extracted). We believe that these ideas could be extended in another way. For example, the relationship between splicing

and RNA modifications may be evaluated in a similar manner based on recent long-read sequencing technology.

(1) On page 2 the authors state:

“unlike previous methods that rely on statistical associations”. This is along the lines of their response as well but we strongly disagree with this assertion. IRAVNet is based on statistical association and the claims it’s not seem to reflect false logic. Yes, IRAVNet can work on a single transcriptome but that’s just because of the way it is defined to filter events. At the end of the day, it is looking for IR reads with variants and the lack of reads without the variants. Seeing enough of those, it calls it an IRV. Is that not a statistical association?? There is definitely not a probabilistic generative model, nor causal modeling, so I’m not sure why the authors make such claims and these should be removed. This view is also reflected in the authors response to my comments which reads:

“....we would like to mention that our approach is not based on statistical associations. Our method is based on the phenomenon that, when variants cause intron retention, those are always found in reads that support intron retention and confirm these facts within the detection pipeline” - again, this argument is inherently flawed. What we have here is an association between detecting IR reads and having these specific variants in those IR reads.

We agree with the Reviewer that the definition of the term “statistical associations” should be more precise. Instead of using “statistical associations,” we add a more specific description as follows:

Another advantage of IRAVNet is that, unlike previous methods that assess the association between genomic variant status and the amount of splicing changes, this pipeline can be run on single transcriptome sequencing data without gathering or specifying a set of data. This makes a systematic screening of public repositories much easier.

Soon after that on the same page the authors write:

“We confirmed that intron retention associated variants (IRAVs) can be detected with high sensitivity and accuracy”.

Actually, the results presented do not support that statement and the basic logic of the method does not support it either. Sensitivity is the amount of TP from those defines as positive. There is no estimate of the total amount of positive (i.e. variants that cause IR) anywhere in the paper. Even the authors themselves seem aware of that. On the following page 3 they state: “Therefore, IRAVNet achieves a certain level of sensitivity and a high rate of precision”. We would agree that the results in the paper, where they compare to SAVNet and IRFinder/MAJIQ, support high precision or low FDR.

Thank you very much. We agree that the sensitivity is hard to measure, and our description was not appropriate. We modified the description as follows:

We confirmed that intron retention associated variants (IRAVs) can be detected with high accuracy as well as a certain degree of sensitivity based on transcriptome sequencing data.

(2) The comparison to IRFinder/MAJIQ is unsatisfactory. The whole point of using such methods is that you could get an estimate of the changes in PSI (IR in this case) in the group with the variant. Instead, the authors choose to lose all the information and just report a z-score between the two groups. Given that there are only IR reads with the variant it is not surprising you would see a significant difference in z-score between the two groups - that is more of a sanity check (an important one!) than any insight into what is being found/reported. For example, running MAJIQ-HET the authors could get a distribution of PSI values in each group with matching statistics. Also, are the p-values for permutations taking into account multiple testing?

We summarized the median difference of PSI (for MAJIQ) or intron retention ratio (for IRFinder) between positive and negative groups for each IRAV. See the newly added Supplementary Figure 7.

We also ran MAJIQ-HET (Vaquero-Garcia, J. et al., bioRxiv, 2021). Please see Supplementary Figure 8 and the “Quantification of intron retention using IRFinder and MAJIQ and comparison of the amount of intron retention with and without IRAVs” in the Method section for the detailed procedure:

We also performed MAJIQ HET to measure the difference between the two groups. For each IRAV, we performed the “majiq heterogen” command on the two groups based on the IRAV mutation status, and executed the “voila tsv” command with “--show-all” option for each VOILA format file. Then, we calculated the difference between the values of columns whose suffixes are “true_median_psi” and “false_median_psi,” and adopted the p-value from the “TNOM” column.

In this validation (Supplementary Figure 5-7), we consider that the multiple comparisons problem is limited. Generally, adjusting p-values for multiple comparisons is most important in the discovery phase, when the number of candidates is typically more than tens of thousands, such as GWAS and differentially expressed gene detection. On the other hand, we evaluated the effect of already detected 66 IRAVs, and z-values and p-values were used for evaluation purposes. Also, since the number of mutations (hypotheses) tested was only 66 in total, it can be inferred that the number of mutations that randomly exceed the significance threshold (0.01 in this case) is minimal.

(3) The comparison to ML based predictions is unsatisfactory and unclear. There is one line in the caption which explains the analysis. Nothing is defined there. What are the FALSE and TRUE groups here? Are the same variants considered?

Please find the description in the Method section (“Comparison to machine learning splicing prediction algorithms”), which had been already included in the previous submission, and the definition of the FALSE and TRUE groups were explained there as “positive” and “negative” groups.

We compared the IRAVNet with existing splicing prediction approaches (MMSplice and SpliceAI) based on machine learning. For each variant at the splice-site regions extracted in the above subsection, we obtained the predictions of splice effects. For MMSplice, we downloaded GTF file from

https://ftp.ebi.ac.uk/pub/databases/gencode/Gencode_human/release_39/gencode.v39.annotation.gtf.gz and performed MMSplice version 2.2.0 using predict_all_table function with the

settings of “pathogenicity=True, splicing_efficiency=True”. We summarized the effect as the maximum across all exons using the max_varEff function when one variant has multiple exons, and the variants whose delta-logit_psi were below -2 were considered to be positive ones. For SpliceAI, we downloaded the pre-calculated annotation (which includes substitutions, 1 base insertions, and 1-4 base deletions within genes) from the Illumina BaseSpace Sequence Hub website, and extracted the score of DS_AL (Delta score (acceptor loss)) or DS_DL (Delta score (donor loss)), and variants whose score were above 0.5 were set to be positive ones. For each approach (IRAVNet, MMSplice, SpliceAI) and for each genomic variant, we divided into positive and negative groups and calculated the Z-value quantified by IRFinder (IRratio and LocalIRratio) as described in the previous subsection.

We modified the label in Supplementary Figure 9b from “TRUE” and “FALSE” to “Positive” and “Negative” to be consistent with the terms used in the Method section. Also, the variants shared by multiple individuals were counted as one in this experiment.

If you take the same variants detected by IRAVNET proximal to splice sites and ran them via these algorithms and compared the correlation of their scores to observed IR or cases where IRFinder/MAJIQ report significant IR, are these predictions so much different/less accurate than the quantification from the RNASeq reads for these variants with IRAVNet? Basically the question is if you used the variant calling from RNASeq for these limited regions and just ran ML prediction algorithms on those, would you get a different result?

Following the Reviewer’s recommendation, we performed GATK, one of the most popular tools for variant calling on the RNA-seq data. According to the recommendation by GATK Team (<https://gatk.broadinstitute.org/hc/en-us/articles/360035531192-RNAseq-short-variant-discovery-SNPs-Indels->), we first processed RNA-seq BAM files and executed variant calling filtering. Then, we extracted variants affecting splice-site proximal regions, followed by normalization and filtering according to the procedure described in the latter part of “Processing Genotype data for GEUVADIS” subsection,

We extracted variants located within the splice-site regions specified by the BED files used in the IRAVNet approach and samples which have matched transcriptome using “bcftools view” and normalized these variants by “bcftools norm.” As performed in IRAVNet workflow, we removed variants whose allele frequencies were more than 0.01 by gnomAD version 3.0 or that were not marked as PASS in the FILTER column, except for the variants identified by IRAVNet. Also, we filtered out those whose differential MaxEntScore by the variants are below 2.5 as performed in the IRAVNet procedure.

In addition, we removed variants that were identified by more than 0.5% of samples (corresponding to AF \geq 0.01). When merging the variants identified by multiple transcriptome data, 380,988 variants remained. This variant set included 63 of 68 IRAVs (92.6%) identified by IRAVNet. On the other hand, the number of variants identified from the matched high coverage whole genome sequencing data was much smaller than that (2,711), which included 66 IRAVs. When comparing the variants with those detected from whole-genome sequencing data from the same individuals, only 293 variants were overlapped (See the Figure below). Considering the maturity of variant detection software from high coverage whole-genome sequencing data, we believe that most mutations detected only by transcriptome are false positives.

There are a variety of possible sources of false positives. One notable phenomenon is that we could observe an excess amount of indels (82.01%) in the variants from transcriptome sequencing data compared to those from whole-genome sequencing data (8.19%). This might be because many split-alignment short reads as well as frequent soft-clipping bases due to misalignment were observed around splice-site regions. These split-alignment and soft-clipping bases may have been incompatible with the local assembly approach of GATK HaplotypeCaller. Several approaches detect variants from RNA-seq, such as RNA-MuTect (Yizhak et al., Science, 2019, DOI: [10.1126/science.aaw072](https://doi.org/10.1126/science.aaw072)). However, this approach avoided variants calling around exon-intron boundaries, possibly because these regions are susceptible to noise due to splicing. **At the very least, it is not easy to accurately identify genomic variants at exon-intron boundaries from transcriptome sequencing data by simply using popular software used for variant calling from genome sequencing data.**

On the other hand, IRAVNet is equipped with a variety of filtering at the mutation detection stage, such as careful examination of the position of the variant in the short reads. Please refer to the “Detection of intron retention associated variants from CRAM format files” subsection in the Method section. There are several essential heuristics to avoid false positives, including but not limited to the following;

Then, we checked the position of the variant in each supporting read and kept variants that are supported by at least 3 different positions (and at least one of which must be inside the 5 bases from the edges).

In addition, verification that the variant is supported specifically to intron retention reads may serve as one circumstantial confirmation of the presence of genomic variation.

Unfortunately, we could not evaluate the effect of intron retention for each variant from RNA-seq because the variant set called from transcriptome sequencing data is far different from those from whole-genome sequencing data, and they are not reliable.

The plots are also unclear (what is the y-axis? Can't see anything about the concentration of points or stats about the distribution in current plots)

The y-axis shows Z-value as described in the method section (please refer to the “Comparison to machine learning splicing prediction algorithms” subsection in the Method section). We changed the size of points (variants) and the overall size of the figure so that the distribution of each point is more visible.

(4) Page 5: “Most of the IRAVs associated with coding regions were predicted to create premature termination codons” - How? This is not defined. 50nt rule from where? How do you know which isoform? The explanatory comment that follows “(premature termination codons are located before the 50bp upstream of the last exon-exon junction)” is not an accurate statement by itself. Clearly there are cases that do not follow this as the authors are well aware. They probably meant to say that they use this as a rule to call PTC (but from which isoform?) - please fix and clarify.

Please find the “Investigating generation of premature termination codon and sensitivity to nonsense mediated-decay” subsection in the Method section, which we had already included in the first submission.

To explore the generation of premature termination codon (PTC) for each IRAV, we chose the longest transcript (defined by RefSeq transcript annotation) harboring the corresponding exon-intron boundary affected by the IRAV. Then, we constructed the transcript with the nucleotide sequence of the retained intron and investigated whether a stop codon occurs before the original termination codon or not. For IRAVs with PTCs, if the PTCs were located before 50 bp upstream of the last exon-exon junction, then those IRAVs were classified into NMD-sensitive, whereas the remaining PTC harboring IRAVs were deemed as NMD-insensitive.

We add the reference to the Method section in the statement of the main section:

(premature termination codons are located before the 50bp upstream of the last exon-exon junction, see Method section for details)

(5) Page 7, section on drug response is unclear and underdeveloped. At the very least, define what is the input data, what metrics are used for drug responses etc.

We thank the Reviewer for pointing this out. Following the Reviewer’s comment, we modified the manuscript as follows:

We also identified IRAVs related to drug response in the same manner as identifying ppIRAVs: we compared positional relationships between variants reported to affect a drug response registered in ClinVar.

We also added the description in the “Definition and classification of putative pathogenic and drug response intron retention associated variants” subsection in the Method section as follows:

The identification and classification of IRAVs putatively affecting drug response were performed in exactly the same way as above, except that variants whose CLNSIG INFO key is “Drug_response” (instead of “Pathogenic,” “Likely_pathogenic,” or “Pathogenic/Likely_pathogenic”) were used as a reference set for positional relationship comparison with IRAVs.

(6) My question about reproducibility was not about subsampling the data (power) but what happens if you only had say 25%, or 50% of the samples and you applied the same

procedure on two subsets of 25% of the data and compared the results - how many of the IRAVNet would have been reproduced?

As discussed previously, IRAVNet has been applied to single transcriptome sequencing data. As such, the results won't be affected by grouping or sample size and exactly the same results will be obtained.

Reviewer 3

The authors have sufficiently responded to my critique. Nice work. Thank you.

I appreciate your very encouraging comments. Thank you very much.

REVIEWERS' COMMENTS

Reviewer #2 (Remarks to the Author):

The authors did a good job addressing remaining comments/concerns. We congratulate them on their work and look forward to seeing it in print.

Response to Reviewers' comments

Reviewer 2

The authors did a good job addressing remaining comments/concerns. We congratulate them on their work and look forward to seeing it in print.

I appreciate your thoughtful suggestions and comments, which improved this manuscript.